

# Overview: The CLoud-Aerosol-Radiation Interaction and Forcing: Year-2017 (CLARIFY-2017) measurement campaign

Jim M. Haywood[1,2], Steven J. Abel[2], Paul A. Barrett[2], Nicolas Bellouin[3], Alan Blyth[4], Keith N. Bower[5], Melissa Brooks[2], Ken Carslaw[4], Haochi Che[5,6], Hugh Coe[7], Michael I. Cotterell[1,8], Ian Crawford[7], Zhiqiang Cui[4], Nicholas Davies[1,9], Beth Dingley[1,5] Paul Field[2,4], Paola Formenti[10], Hamish Gordon[4,11], Martin de Graaf[12], Ross Herbert[3], Ben Johnson[2], Anthony C. Jones[1,2], Justin M. Langridge[2], Florent Malavelle[1,2], Daniel G. Partridge[1], Fanny Peers[1], Jens Redemann[13], Philip Stier[5], Kate Szpek[2], Jonathan W. Taylor[7], Duncan Watson-Parris[5], Robert Wood[14], Huihui Wu[7], Paquita Zuidema[15].

[1] College of Engineering, Mathematics and Physical Science, University of Exeter, Exeter, UK, EX4 4QE.
[2] Met Office, Exeter, UK, EX1 3PB.
[3] Dept. of Meteorology, University of Reading, UK, RG6 6BB.
[4] School of Earth and Environment, University of Leeds, UK, LS2 9JT.
[5] Atmospheric, Oceanic and Planetary Physics, Department of Physics, University of Oxford, UK, OX1 3PU.
[6] Now at Tel Aviv University, Tel Aviv, Israel.
[7] Department of Earth and Environmental Sciences, University of Manchester, M13 9PL, UK.
[8] Now at School of Chemistry, University of Bristol, Bristol, UK, BS8 1TS.
[9] Now at Haseltine Lake Kempner, Bristol, UK, BS1 6HU.
[10] LISA, UMR CNRS 7583, Université Paris-Est-Créteil, Université de Paris, Institut Pierre Simon Laplace (IPSL), Créteil, France.
[11] Now at Engineering Research Accelerator, Carnegie Mellon University, Pittsburgh, Pennsylvania, USA.
[12] KNMI, Netherlands.
[13] University of Oklahoma, Oklahoma, USA.
[14] University of Washington, Washington, USA.
[15] Rosenstiel School of Marine and Atmospheric Science, University of Miami, Florida, USA.

*Correspondence to*: Jim M. Haywood (J.M.Haywood@exeter.ac.uk)

**Abstract.** The representation of clouds, aerosols and cloud-aerosol-radiation impacts remain some of the largest uncertainties in climate change, limiting our ability to accurately reconstruct and predict future climate. The south-east Atlantic is a region where high atmospheric aerosol loadings and semi-permanent stratocumulus clouds are co-located, providing a natural laboratory for studying the full range of aerosol-radiation and aerosol-cloud interactions and their perturbations of the Earth's radiation budget. While satellite measurements have provided some useful insights into aerosol-radiation and aerosol-cloud interactions over the region, these observations do not have the spatial and temporal resolution, nor the required level of precision to allow for a process level assessment. Detailed measurements from high spatial and temporal resolution airborne atmospheric measurements in the region are very sparse, limiting their use in assessing the performance of aerosol modelling in numerical weather prediction and climate models. CLARIFY-2017 was a major consortium programme consisting of 5 principal UK universities with project partners from the UK Met Office and European and USA-based universities and research centres involved in the complementary ORACLES, LASIC and AEROCLO-sA projects. The aims of CLARIFY-2017 were four-fold; 1) to improve the representation and reduce uncertainty in model estimates of the direct, semi-direct and indirect radiative effect of absorbing biomass burning aerosols; 2) improve our knowledge and representation of the processes determining stratocumulus cloud microphysical and radiative properties and their transition to cumulus regimes; 3) challenge, validate and improve satellite retrievals of cloud and aerosol properties and their radiative impacts; 4) improve numerical models of cloud and aerosol and their impacts on radiation, weather and climate. This paper describes the modelling and measurement strategies central to the CLARIFY-2017 deployment of the FAAM BAe146 instrumented aircraft campaign, summarises the flight objectives and flight patterns, and highlights some key results from our initial analyses.



## 1. Introduction and Rationale

The interaction of clouds, aerosols and radiation are highlighted as key climate uncertainties in the recent Intergovernmental Panel on Climate Change (IPCC) assessment report (Boucher *et al*., 2013). Aerosol-radiation interactions stem from direct scattering and absorption of solar and terrestrial radiation by aerosols, thereby changing the planetary albedo. Aerosol-cloud interactions, also termed indirect effects, arise from aerosols acting as cloud condensation nuclei (CCN) in warm clouds. An increase in the number of activated CCN for fixed liquid water path translates into larger concentrations of smaller cloud droplets, increasing cloud albedo (Twomey, 1974). Both aerosol-radiation and aerosol-cloud interactions trigger fast adjustments to the profiles of temperature, moisture, and cloud water content, which ultimately may affect cloud formation and precipitation rates and cloud lifetime (e.g. Albrecht, 1989; Pincus and Baker, 1994; Johnson *et al*., 2004). The quantification of interactions in the cloud-aerosol-radiation system remains elusive. The recent IPCC report (Boucher *et al*., 2013) stresses that aerosol climate impacts remain the largest uncertainty in driving climate change, with a global mean effective forcing of -0.50 $\pm$ 0.40 W m$^{-2}$ for the aerosol-radiation-interaction and in the range of 0.0 to -0.9 W m$^{-2}$ for the aerosol-cloud-interaction thereby counter-balancing a significant, but poorly constrained, fraction of greenhouse gas-induced global warming which is estimated as +2.8 $\pm$ 0.3 W m$^{-2}$ (Myhre *et al*., 2013a). This uncertainty impacts our ability to attribute climate change, to quantify climate sensitivity, and therefore to improve the accuracy of future climate change projections. In regions with strong anthropogenic influences, aerosol radiative forcings are an order of magnitude larger than their global mean values, limiting our ability to provide reliable regional climate projections.

Biomass burning smoke aerosol (BBA) consists of complex organic carbon compounds mixed with black carbon and inorganic species such as nitrate and sulfate. Black carbon is a strong absorber of sunlight (e.g. Shindell *et al*., 2012; Bond *et al*., 2013) and certain organic compounds (so-called 'brown carbon') also absorb sunlight, particularly at shorter UV wavelengths (e.g. Andreae and Gelencsér, 2006). BBA is an important component of anthropogenic aerosol and is produced from fires associated with deforestation, savannah burning, agricultural waste, and domestic biofuels with global emissions estimated to have increased by 25% since pre-industrial times (Lamarque *et al*., 2010). The African continent is the largest global source of BBA, currently contributing around 2-29 Tg[C] year$^{-1}$ (with [C] indicating this emission rate corresponds to that of carbon) or 50% of global emissions (e.g. van der Werf *et al*., 2010; Bond *et al*., 2013). The meteorological transport of BBA over southern Africa during the dry season is dominated by an anticyclonic circulation with westward transport on the northern periphery and eastward transport on the southern periphery (Adebiyi and Zuidema, 2016; Swap *et al*., 2002; Garstang *et al*., 1996). Over the continent, vertical mixing is inhibited by stable layers at the top of the continental boundary layer and by the main subsidence inversion (around 5 – 6 km above sea level, ASL) (Harrison, 1993; Garstang *et al*., 1996).

\*\*\*Insert Figure 1 here\*\*\*

Over the South East (SE) Atlantic, the BBA in the residual continental boundary layer (CBL) over-rides the marine boundary layer (MBL) where low sea-surface temperatures and large-scale subsidence give rise to persistent stratocumulus cloud, as evidenced in Figure 1 that shows the climatology of cloud fraction and aerosol optical depth (AOD). A large temperature inversion may inhibit mixing between the BBA in the elevated residual CBL and the marine boundary layer which, in turn, may limit the interaction with the clouds. However, prior to CLARIFY-2017, the degree of aerosol-cloud interaction was highly uncertain, highlighting the need for comprehensive *in situ* measurements.

While developing the scientific rationale for CLARIFY-2017, it became obvious that interest in aerosol-cloud and aerosol-radiation interactions in the SE Atlantic region extended well beyond the UK community. Not only were additional European project partners entrained into CLARIFY-2017, but synergistic measurement campaigns planned by other multi-national research groups were also developed. Of specific complementary synergy were:-





LASIC (Layered Atlantic Smoke Interactions with Clouds) which deployed a large suite of surface-based observations via the Atmospheric

Radiation Measurement (ARM) Mobile Facility (AMF; https://www.arm.gov/capabilities/observatories/amf) to Ascension Island between

July 2016 – October 2017 (Zuidema *et al.*, 2018a).

ORACLES (ObseRvations of Aerosols above CLouds and their intEractionS which deployed the high altitude ER2 and heavily instrumented

P3 aircraft to Walvis Bay, Namibia in September 2016 and the P3 alone to São Tomé in August, 2017 and October, 2018 (Redemann *et al.*,

2020).

AEROCLO-sA (AErosol, RadiatiOn and CLOuds in southern Africa) which deployed a surface mobile platform and the instrumented

French Falcon 20 environmental research aircraft of Safire in Henties Bay and Walvis Bay, respectively, in 2017 (Formenti *et al.*, 2019).

All of these measurement campaigns comprised major deployments of research assets to the South Atlantic Region during 2017 (Zuidema

*et al.*, 2016). The location of these campaigns is summarised on Figure 1. The scientific steering committees of the four synergistic projects

frequently included members from the other projects. Planning teams from CLARIFY-2017, ORACLES, LASIC and AEROCLO-sA kept

in close contact during their planning, deployment and analyses phases which led to many benefits such as forecast model sharing, joint

special sessions at the EGU and AGU, and a mutual physically located workshop in Paris and a virtual workshop in Miami (owing to the

Covid-19 travel restrictions) dedicated to cross-campaign collaboration. An inter-comparison flight was performed between the FAAM

BAe146 aircraft and the NASA P3 aircraft when both were operating from Ascension Island during 2017 allowing an assessment and inter-

comparison of the performance characteristics of the aircraft instruments (Barrett *et al.*, 2020a).

We acknowledge here that the results from CLARIFY-2017, ORACLES, LASIC and AEROCLO-sA campaigns are already starting to

appear in the scientific literature, particularly as part of this thematic special issue. This section lays out the original motivation of the

CLARIFY-2017 campaign prior to intensive modelling and observations. We defer a discussion of these various studies to later sections of

this work.

### 1.1 Aerosol-Radiation Interactions (ARI)

On a global mean basis BBA is estimated to exert a neutral direct radiative forcing of -0.1 to +0.1 W m$^{-2}$ (Boucher *et al.*, 2013). Even the

sign of the global mean direct radiative forcing is in doubt because the single scattering albedo (SSA, the ratio of optical attenuation

coefficients for scattering and extinction) of BBAs is close to the balance point between net reflection and net absorption of sunlight (e.g.

Haywood and Shine, 1995). However, regionally, BBA plays a far more important role: nowhere is the uncertainty in the direct radiative

effect and forcing more apparent than over the SE Atlantic than during the August-September dry season (Figure 2).


\*\*\*Insert Figure 2 here \*\*\*

Figure 2 shows the 'direct' radiative effect derived from models participating in AEROCOM (Myhre *et al.*, 2013b; Stier *et al.*, 2013)

indicating a regional hotspot for BBA forcing over the SE Atlantic but with significant uncertainty because BBA can exist either above the

stratocumulus decreasing the planetary albedo or above open ocean where it increases the planetary albedo. To accurately model the aerosol

direct effect, models need to represent all of the following correctly: the magnitude and geographic distribution of the AOD, the wavelength

dependent SSA, the BBA vertical profile, the geographic distribution of the cloud, the cloud fraction, the cloud liquid water content, the

cloud droplet effective radii, and the cloud vertical profile (Keil and Haywood, 2003; Abel *et al.*, 2005; Samset *et al.*, 2013; Stier *et al.*,

2013). At a more detailed aerosol process level, we need to understand the optical properties of black carbon, organic carbon and inorganic

compounds as a function of mixing state and how these properties vary as a function of altitude, relative humidity and as a function of aging

from emission to deposition.





Another implication of BBA overlying cloud is that satellite retrievals of cloud that rely on visible wavelengths are generally biased low in cloud optical depth (COD) and effective radius (e.g. Hsu *et al*., 2003; Haywood *et al*., 2004, Wilcox and Platnick, 2009) with implications for remotely sensed correlative studies of aerosol-cloud interactions (Quaas *et al*., 2008). Recently, de Graaf *et al*, (2012) used high spectral

resolution satellite data to show that the direct radiative effect of BBA over clouds in the SE Atlantic region could be stronger than +130 W $m^{-2}$ instantaneously and +23 W $m^{-2}$ in the monthly mean. These values are far stronger than those diagnosed in climate models which reach only +50 W $m^{-2}$ instantaneously (e.g. de Graaf *et al*., 2014), suggesting that models misrepresent at least one key parameter noted above.

A further aerosol-radiation interaction occurs as a fast adjustment to the direct effect and is called the semi-direct effect (SDE), whereby the heating of the absorbing BBA layer and the reduction in surface temperature modify the atmospheric stability, surface fluxes, clouds

and hence radiation. Satellite observations over southern Atlantic stratocumulus have shown a thickening of cloud underlying BBA (Wilcox, 2012, Costantino and Breon, 2013) which could be a result of heating of the above cloud column intensifying the cloud top inversion and reducing entrainment. Wilcox (2012) estimated that this produced a negative radiative effect that compensated for 60% of the above cloud positive direct effect. Large eddy model (LEM) simulations have been used to explore the detailed mechanisms of the semi-direct effect (e.g. Johnson *et al*, 2004, Hill and Dobbie, 2008) although they typically have relatively small domain sizes and therefore cannot account

for the impact of aerosol in modifying synoptic scale circulations. Global modelling studies are able to represent impacts on synoptic and regional scale dynamics and circulation patterns (e.g. Allen and Sherwood, 2010; Randles and Ramaswamy, 2010) but are unable to represent the detailed process level mechanisms captured by LEMs. Studies in LEMs and global climate models have emphasised the importance of the vertical profile of aerosol and the degree of absorption (Johnson, 2004, Randles and Ramaswamy, 2010, Samset *et al*., 2013). Randles and Ramaswamy (2010) and Allen and Sherwood (2010) document the response to the semi-direct effect via atmospheric

impacts on stabilisation, reduced surface fluxes and subsequent evolution of the modelled dynamical impacts. Climate models need to parameterise many of the mechanisms by which the semi-direct effect operates and the climate response is likely to be sensitive to the details of the parameterisation. Johnson (2004) found the semi-direct effect to be 5 times smaller in global scale models compared to LEMs although these results are challenged by Allen and Sherwood (2010). In addition, internal variability masks local semi-direct effects, severely decreasing the statistical significance in previous studies of modelled semi-direct effects (e.g. Ghan *et al*., 2012) and our ability to

assess their fidelity.

### 1.2 Aerosol-Cloud Interactions (ACI)

Despite considerable advances in modelling clouds using models of different resolutions, considerable uncertainties remain in modelling even the relatively simple cases of stratocumulus owing to uncertainties in precipitation, decoupling, moisture budgets and entrainment. Unsurprisingly, climate models show considerable inter-model biases in cloud fraction, liquid water path, effective radius and COD when

compared against satellite observations leading to large discrepancies in the solar fluxes and hence the energy absorbed by the ocean in the region (e.g. Bodas-Salcedo *et al*., 2014).

Aerosol-cloud interactions, or 'indirect effects', remain one of the most elusive but key parameters in climate prediction (Stevens and Feingold, 2009; Boucher *et al*., 2013). For stratocumulus, the effect of increased CCN leading to cloud brightening can be modulated by changes in precipitation and subsequent changes to cloud water amounts through entrainment processes (e.g. Ackerman *et al*. 2004).

Satellite-borne lidar studies of aerosol-cloud interactions in the region emphasise the critical role of the vertical profile of aerosol and cloud (Costantino and Breon, 2013) and the relative position of the two to each other (Chand et al. 2009). However, it is difficult to fully discern the level of interaction between clouds and aerosols because of the sensitivity of lidars in the free troposphere (Watson-Parris *et al*., 2018) and the attenuating effects of a thick layer of aerosols overlying clouds. Global bulk aerosol models and empirical representations of aerosol indirect effects are being replaced with microphysical aerosol models such as ECHAM5-HAM (Stier *et al*., 2005) and GLOMAP (also

known as UKCA-mode) (e.g. Mann *et al*., 2010; Bellouin *et al*., 2013) and more explicit representation of cloud and precipitation processes





(Hill *et al*., 2015; Grosvenor *et al*., 2017) have also been developed. Such schemes require extensive evaluation which are often achieved through multi-model intercomparison studies (e.g Quaas *et al*., 2009) and comparison to observations. The spatial resolution of global numerical weather prediction (NWP) and climate models (typically 10~100 km) is widely recognised as inadequate for investigating essential aerosol-cloud interaction processes at the cloud scale (~10 m; Lebo *et al*., 2017). Thus, relationships between sub-grid-scale
variables such as cloud updraft velocity and entrainment from LEMs and their link to large scale boundary layer variables are being sought, but, while promising, are far from well established (e.g. Golaz *et al*., 2011; Malavelle *et al*., 2014). Simulations with HadGEM2-Coupled Large-scale Aerosol Scheme for Studies In Climate (CLASSIC) aerosol scheme suggest that, while BBA interaction with cloud may be limited by vertical stratification, it does enter the MBL and interact with cloud producing a strong indirect effect in the region (Figure 3a). However, the more sophisticated GLOMAP-MODE two-moment scheme leads to a much reduced aerosol indirect effect because an
increase in aerosol mass does not necessarily lead to an increase in the aerosol number or CCN as the aerosol size distribution will tend to shift to larger sizes as more volatile organic precursors condense upon pre-existing aerosol particles (Figure 3b). The over-strong aerosol-cloud-interaction in CLASSIC compared to GLOMAP-MODE has been noted in other studies that have used satellite retrievals to assess their validity (e.g. Malavelle *et al*., 2017).

\*\*\*Insert Figure 3\*\*\*

An assessment of parametric uncertainty in the GLOMAP-MODE global model driven by ECMWF meteorology and observed low-level clouds (Lee *et al*., 2013) showed that BBA particles are one of the largest sources of uncertainty in CCN at cloud base. However, Lee *et al*. (2013) did not assess the effect of uncertainties in the physical model, which control the extent to which BBA and clouds mix, nor structural model uncertainties.
The stratocumulus decks of the SE Atlantic have been linked via global teleconnections to precipitation anomalies in Brazilian rainfall; SE Atlantic stratocumulus that is too bright can lead to precipitation deficits in the Norde-Est and Amazonian regions (Milton and Earnshaw, 2007; Jones *et al*., 2009). Similarly, Atlantic sea-surface temperature gradients and the hemispherical asymmetry in the energy balance are strongly impacted by SE Atlantic stratocumulus (Jones and Haywood, 2012; Stephens *et al*., 2016) influencing the position of the Inter-Tropical Convergence Zone (ITCZ), and hence the African and Asian monsoon.

**1.3 Previous measurements in the region, and advances since then.**

The last major international measurement campaign investigating biomass burning in Southern Africa was the Southern AFricAn Regional science Initiative in 2000 (SAFARI-2000). The SAFARI-2000 dry-season intensive campaign focussed on the emissions, transport and transformation of BBA plumes and the validation of satellite remote sensing retrievals of aerosol and cloud from the Terra satellite (Swap *et al*., 2002). The majority of investigations over the SE Atlantic were basic aerosol microphysics and cloud-free radiative impact studies
(Haywood *et al*., 2003, Keil and Haywood, 2003, Osborne *et al*., 2004, Magi et al., 2008). Since SAFARI-2000, significant advances in airborne measurement of BC (e.g. Schwarz *et al*., 2008; McMeeking *et al*., 2011); organic and inorganic aerosol compounds (Morgan *et al*., 2010) and aerosol physical properties have occurred. In addition, improvements in the accuracy and sensitivity of measurements of aerosol optical properties, notably absorption (e,g. Sedlacek and Lee, 2007; Lack *et al*., 2008) have been made. Airborne lidar instrumentation and retrievals allow concurrent mapping of vertical distributions of aerosols above clouds (e.g. Marenco *et al*., 2011). An
extensive set of measurements of stratocumulus clouds has been performed during VOCALS off the Pacific coast of South America (Wood *et al*., 2011) with one of the foci being aerosol-cloud interactions (e.g. Yang *et al*., 2011; Painemal and Zuidema, 2013). However, the aerosol composition, sources and interaction with the clouds in the VOCALS region are very different to those over the SE Atlantic which is dominated by relatively strongly absorbing biomass burning aerosol (e.g. Haywood *et al*., 2003). Model capabilities have also improved. At the time of SAFARI-2000, aerosol modelling was in its infancy with only two global chemical transport models reporting the direct





radiative forcing and cloud-albedo indirect forcing of BBA in the IPCC report (Ramaswamy *et al*, 2001). Since 2000, the focus for aerosol-radiation interactions has shifted to areas where model results diverge (e.g. SE Atlantic, see Figure 2 and Figure 3). Global aerosol microphysics models have also been developed and are coupled to climate models and to cloud models at high resolution. Aerosol-cloud interactions are now studied at scales ranging from LEMs with resolutions of a few meters, through cloud resolving models, and limited area numerical weather prediction models to global models with resolutions of ~100 km. New approaches to understand sources of aerosol

uncertainty have also been developed (Lee *et al*., 2013). However, high quality validation data in the SE Atlantic with which to challenge the global and cloud resolving models is almost entirely lacking.

### 1.4 Key Aims and Objectives.

With the rationale as described above, CLARIFY-2017 aimed to use the natural laboratory of the SE Atlantic to improve the representation of BBAs and clouds in models of a range of scales, increase the fidelity of aerosol-radiation and aerosol-cloud interaction processes and

cloud representation, and their impacts on local, regional and global weather and climate. Experience suggested that these objectives were best achieved by conducting an intensive airborne field campaign with supporting surface and satellite measurements. The measurements were used to challenge and develop improved models at different spatial scales from the cloud scale to the global scale that couple aerosols, clouds and radiation.

Specific key objectives of CLARIFY-2017 were:

Key Objective 1: Measure and understand the physical, chemical, optical and radiative properties of BBAs in the SE Atlantic region.

Key Objective 2: Understand, evaluate and improve the physical properties of the SE Atlantic stratocumulus clouds and their environment in a range of models.

Key Objective 3: Evaluate and improve the representation of BBA-radiation interactions over the SE Atlantic when clouds are absent/present at a range of model scales and resolutions.

Key Objective 4: Evaluate and improve the representation of BBA-cloud interactions over the SE Atlantic at a range of model scales and resolutions.

The purpose of this work is to describe the deployment strategy (section 2), the aircraft and surface-based instrumentation (section 3), the flight patterns used to deliver specific objectives (section 4), a summary of the flights performed (section 5), and to signpost certain key initial results (section 6). Conclusions are presented in section 7.

## 2. Deployment Strategy

CLARIFY was originally scheduled to operate from Walvis Bay, Namibia in August-September 2016. August-September was chosen as an optimal operating window via analysis of multi-year satellite analyses and surface-based sun-photometer observations (Adebiyi *et al*., 2015; multi-year means presented in detail in Redemann *et al*., 2020). There was also evidence that suitably high aerosol loadings had been encountered during this period during aircraft operations with the UK's instrumented C-130 aircraft and the University of Washington's

CV-580 aircraft during SAFARI-2000 (Haywood *et al*., 2003; Hobbs, 2003; Osborne *et al*., 2004). However, operating permissions were not forthcoming although permission was eventually given for operations in 2016 for the ORACLES campaign, and in 2017 for the AEROCLO-sA campaign. Given the ORACLES and AEROCLO-sA deployments were based from continental Africa relatively close to the sources of biomass burning, additional merits were envisaged locating CLARIFY-2017 operations at a downstream location enabling very aged BBA to be sampled. These factors, together with the deployment of the AMF under the LASIC proposal led to the decision to

relocate to Ascension Island and delay deployment until August-September 2017, given that the biomass plume and underlying



stratocumulus decks could be accessed from Ascension Island. The deployment was given the full support of UK's Joint Forces Command which aided the logistics of deployment.

To ensure that model and observational products were readily available and that the scientists were familiar with the likely meteorological, cloud and aerosol conditions, dry-run periods were established one year ahead of deployment (July-Sep 2016) and one month prior to

deployment (Jul-Aug 2017). The benefits of holding a dry-run during Aug-Sept 2016 were enhanced by the in-field operations and the associated modelling support of the ORACLES campaign.

Tools for flight planning included global and regional model forecasts, satellite analyses and data feeds from surface-based instrumentation. Global modelling efforts for 2016 included the following models: ECMWF, UK Met Office, the Global Forecasting System of NCEP and the GEOS-5 model (see Redemann *et al.*, 2020), all of which provided their standard meteorological variables such as cloud fraction, cloud

liquid water, boundary layer depth etc. In addition, the Met Office developed a bespoke three-component aerosol system for use in its development version of the global NWP model (Walters *et al.*, 2011) which ran at around 15 km spatial resolution. The aerosol model was essentially a derivative of the CLASSIC scheme (e.g. Bellouin *et al.*, 2011) in which aerosols are modelled as externally mixed. The three components that were chosen were i) sulfate with emissions from industrial pollution and dimethyl sulphide (DMS), ii) a simplified two-bin mineral dust scheme based on Woodward (2001) with interactive emissions and data assimilation from MODIS Aqua, and iii)

'carbonaceous aerosols' with real-time fire emissions from fossil fuel, biofuel and real-time fire emissions (Global Fire Assimilation System (GFAS); Kaiser *et al.*, 2012) combined into one tracer. The three aerosol components were chosen as a compromise because the model was also used in the South West Asian Aerosol Monsoon Interaction (SWAAMI) campaign (e.g. Brooks *et al.*, 2019) and in the Dynamics–aerosol–chemistry–cloud interactions in West Africa (DACCIWA) project (Knippertz *et al.*, 2015). This 3-component aerosol model is known as the CLASSIC-Lumped (CLUMP) model owing to the emissions being lumped into source terms for the three aerosol components.

While the limitations of such single moment schemes are recognised, the primary purpose of the scheme was to locate the aircraft in approximately the right place at the right time. Note that, with the exception of the impacts of mineral dust that are included in the operational model, aerosol-radiation-interactions and aerosol-cloud-interactions are explicitly turned off in CLUMP so that the dynamical evolution of the developmental model is identical to the operational model. Examples of some of the bespoke products are shown in Figure 4 and Figure 5.


***Insert Fig 4 ***

Figure 4a shows the expected pattern of aerosol optical depth (at a wavelength of 550nm), although it is to the north of the seasonally averaged August-October AODs (Figure 1) owing to the more northerly location of biomass burning at this time of year. The CODs shown

in Figure 4b show the level of detail that is possible in a high-resolution numerical weather prediction model. Figure 4c shows the above cloud aerosol optical depth (ACAOD) that is diagnosed from the model together with transects on radials originating from Ascension Island that were routinely analysed during the dry-run and deployment periods.

*** Inset Fig 5***


Figure 5 shows further bespoke model products along the 70° and 130° transects shown in Figure 4c. The 70° radial heads into the heart of the biomass burning plume and suggests a very different degree of vertical mixing when compared to that at 130°. The 70° transect suggests that the carbonaceous aerosol originating from biomass burning is mixed throughout the boundary layer with around 75% on average of carbonaceous aerosol residing above cloud and the remaining 25% being contained within the MBL. This leads to classifications of cloud



that are generally polluted (arbitrary threshold of 3 µg kg$^{-1}$of BBA) or very polluted (arbitrary threshold of 10 µg kg$^{-1}$ of BBA) within the modelling framework.

The 130° transect is very different with the carbonaceous aerosol almost entirely overlying cloud. This leads to classifications of cloud that are either 'close' to interacting with cloud (when the aerosol base is within 200 m of the cloud) or 'clean' when there is little in the way of biomass burning present to the south of the region. Of course, the utility of the model as a forecast tool depends on its ability to accurately

represent the details of mixing of the BBA down from the residual CBL into the MBL; we will show that the model is indeed capable of capturing these features in Section 6. In addition to the UK Met Office 15 km resolution model, two other global aerosol models were available. ECMWF were a project partner on the CLARIFY project and provided ECMWF-based model forecasts from Copernicus Atmosphere Monitoring Service (CAMS; https://atmosphere.copernicus.eu/) and the NASA-based GEOS5 (https://gmao.gsfc.nasa.gov/GEOS/) that was run in support of the ORACLES programme (Redemann *et al*., 2020).

Limited area NWP forecast models were also utilised with a horizontal grid-spacing of 4 km and a domain of around 2000km x 2000km with boundary conditions provided by the global NWP model; this model did not include aerosol transport but provided even higher resolution cloud products. Regional models that did include aerosol were also run in support of ORACLES e.g. WRF-Chem at 36 km resolution and full-chemistry and WRF-Aerosol Aware Microphysics at 12 km resolution (Saide *et al*., 2016), the latter model being the primary ORACLES aerosol forecast tool. The formulation and resolution of the WRF-Aerosol Aware Microphysics simulations are similar

to those of the Met Office NWP-CLUMP model although the NWP-CLUMP model had aerosol-radiation and aerosol-cloud interactions disabled. Output data products from these models were also made available to the CLARIFY team (Redemann *et al*., 2020).

Satellite products provided another important tool for planning aircraft flights. MODIS was used to provide one-day old observations of aerosol optical depth in cloud-free regions while the geostationary SEVIRI instrument was used for now-casting cloud conditions with images of cloud conditions being periodically relayed to the FAAM aircraft throughout the flight. Before the dry-runs and deployment, a

register of the timing and track of overpasses from polar orbiting satellites (e.g. Terra, Aqua) that were in the vicinity of Ascension Island was made taking care to exclude areas where the satellite was influenced by sun-glint. This allowed scientists to decide on the relative priorities of flights. In the case that forecasted AOD and cloud-cover conditions were expected to be consistent for several days, priority was given to those days with local satellite overpasses so the aircraft measurements could provide data for satellite validation.

Further information from ground-based instrumentation was utilised for now-casting. This included information on the aerosol optical depth

from two Cimels sun-photometers based at the AERONET and AMF sites, a hand-held Microtops sunphometer based at the operations centre in Georgetown, and a LEOSPHERE depolarizing lidar, operated by KNMI at the airfield. This combination of equipment allowed an assessment of the aerosol loadings and the vertical distribution of aerosol relative to cloud prior to aircraft take-off. Further details of this instrumentation together with the aircraft instrumentation are given in Section 3.

### 3 Aircraft and surface-based instrumentation

The BAe146 FAAM aircraft is the UK's NERC-funded atmospheric research aircraft and is part-funded by the UK's Met Office. It has the largest payload of any European atmospheric research aircraft, capable of carrying 3 crew, 18 scientists and a total scientific payload of up to 4000 kg for a distance of 3700 km with a ceiling of 35,000 feet and has a typical science airspeed of 110 m s$^{-1}$. The endurance of the BAe146 aircraft is typically up to 6 hours depending on the scientific payload, the flight patterns, ambient meteorological conditions and the proximity of diversion airports.

The aircraft instrumentation used in this configuration is an enhanced version of that used in previous aerosol/radiation campaigns such as DABEX and GERBILS (e.g. Haywood *et al*., 2008, 2011) and is broken down into sub-sets corresponding to aerosol microphysics, aerosol



composition and optical properties, cloud physics, radiation and remote sensing, trace gas chemistry and thermodynamics is summarised in Table 1.

***Insert Table 1***

The instrumentation was chosen to provide an optimal instrumentational fit to meet the key objectives, while keeping down the operational weight of the aircraft to maintain a reasonable range.

In addition to the aircraft instrumentation and the sun-photometer and lidar deployed at the airfield, the deployment to Ascension Island
benefitted from the synergistic deployment of the AMF to Ascension Island. The AMF was located on a more remote windward side of the island, to avoid local aerosol sources, at a site approximately 300 m (1000 ft) above sea level (Zuidema *et al*., 2016; 2018). The deployment spanned July 2016 – October 2017 and thus captured two distinct biomass burning seasons. The FAAM aircraft made several fly-pasts of the AMF site at 1000 ft ASL offset by approximately 2 km to the east so that it was operating at the same altitude thus allowing a comparison of aerosol, trace gas and radiation measurements. A new HANDIX Portable Optical Particle Counter (POPS) was also operated at the AMF
by the University of Exeter for the duration of the FAAM deployment to help provide a long-term characterisation of the instrument. Standard meteorological measurements were also made by the Met Office located at the airfield including precipitation measurements. A long-standing standard Cimel sun-photometer has also been operational on Ascension Island as part of the AERONET network since 1998.

**4 Flight patterns for the objectives**

Because the aircraft was operating from Ascension Island where there are no diversion airports, island holding restrictions were in place
resulting in a reduced operating duration of around 3.5-4hours. Owing to these restrictions, extended operations at distances far from Ascension Island were curtailed. However, owing to the significant cooperation of the RAF, USAF, ATC, fire-crew and ground-crew, the aircraft was able to operate for two flights per day if required from 09:00-12:30 followed by re-fuel and flying 14:00-17:30 Monday-Friday. No flights were permitted on Saturday afternoons taking account of other air traffic utilising the airstrip and Sunday was classed as a hard-down day with no flying permitted to provide a scheduled rest-day. Scientific outreach showcasing the aircraft and our science was via
guided tours of the aircraft and talks on the scientific research being performed to the general population of Ascension Island (approximately 1/3 of the Island's entire civilian population were present).

Depending on the aerosol and cloud conditions determined from forecast products, satellite retrievals and ground-based observation data, the FAAM aircraft flights were designed to characterise the main aerosol and cloud state in clean and polluted conditions and to study properties and processes rather than to build a spatially and temporally representative mapping of the region (see Redemann *et al*., 2020 for
ORACLES flight plans for building such a representative mapping). This strategy to flight planning ensured suitable data sets were collected to facilitate meeting the key objectives described in Sect. 1.4.

A series of pre-determined, but flexible, flight patterns were developed (e.g. Figure 6). Each flight pattern was made up of a series of manoeuvres including 'straight and level runs (SLRs)' (denoted #1, #2, #3, and #5 in Figure 6), 'profiles' (denoted #6 in Figure 6), 'saw-tooths (denoted #4 in the cloudy-flight schematic of Figure 6)' and 'orbits' (denoted #4 in the cloud-free schematic of Figure 6). SLRs of
differing duration were made at constant pressure levels. Profiles were typically made at a constant rate of descent/ascent of 1000 ft per minute (although 500 ft/minute was typical at the lowest levels), while saw-tooths were frequently used from cloud-top to cloud-base to characterise clouds. Orbits in conjunction with the SWS instrument are flown at high angles of bank (typically 60°), take less than 2 minutes to complete, and allow measurements that are analogous to Cimel almucantar scans (Osborne *et al*, 2008, 2011).





***Insert Figure 6***

A manoeuvre carried out when the skies were predominantly cloud-free while the aircraft was on the ground consisted of a 'pirouette'; rotating the aircraft through 360 degrees over a period of around 2 minutes while the aircraft was on the Ascension Island runway or apron. This allowed two separate measurements to be made. Firstly, levelling corrections for the Eppley BBR and SHIMS instruments (Table 1)

could be performed from these manoeuvres and any impacts of dome degradation via aerosol impaction on the front faces of the BBR and SHIMS domes could be assessed by examining pre- and post-flight data (Barrett *et al*., 2020a). Secondly, by setting the SWS instruments viewing geometry to match the solar zenith angle (or the solar angle plus 10 degrees), the SWS instrument effectively made almucantar scans analogous to those made by Cimel sun-photometers where the radiance is mapped out as a function of the scattering angle. By setting the SWS viewing geometry to the solar zenith angle plus 10°, the range of scattering angles sampled was from 10° to twice the solar zenith

angle plus 20°.

Flight patterns for aerosol characterisation generally consisted of either SLRs through the BBA layer or vertical profiles/sawtooths to constrain their vertical distribution in the atmospheric column. Because both the radiation and cloud sorties described below involved many measurements of aerosol, specific flight patterns focussing solely on aerosol characterisation were not performed; aerosol characterisation was implicit within the other sorties and mainly used a combination of SLRs and vertical profiles.

**4.1 Flight patterns for radiation objectives**

An example of the flight patterns performed for determining the radiative effects of BBA in cloud-free and cloudy skies is shown in Figure 6 based on preconceived ideas of what we would expect based on prior experience from SAFARI-2000 (Haywood *et al*., 2003).

The patterns shown in Figure 6 were typically orientated so that the straight and level runs (SLRs) and profiles avoided running within 30 degrees of the into-sun heading. This is to avoid making radiative transfer measurements where aerosol may have been impacted on the

front face of the Eppley BBRs and SHIMS instruments which could lead to a reduction in measured irradiance. Owing to the variability of cloud, the order of the runs was typically changed during CLARIFY-2017 so that the high level SLR leg was followed by a reciprocal turn and profile descent followed by reciprocal-turn and SLR just above cloud top. This ensured the minimum length of time had elapsed between the two legs to minimise differences caused by changes in cloud fields below the aircraft (e.g. Peers *et al*., 2019, 2020). Because of the shape of this sequence, the pattern is known as a 'Z-pattern'.

Radiometric measurements above and below the BBA characterised broadband and spectral irradiances and radiances, provided aerosol vertical distribution from lidar and enabled sea-surface reflectance characterisation. Profiles through the BBA characterised the aerosol extinction and absorption coefficient from the EXtinction, Scattering and Absorption of Light for AirBorne Aerosol Research (EXSCALABAR) instrument and hence the aerosol optical depth and aerosol absorption optical depth. When BBA overlies cloud, SLRs above and below BBA provided remotely sensed estimates of cloud-top droplet effective radius and LWP from solar and microwave

instrumentation.

**4.2 Flight patterns for cloud-characterisation objectives**

Flight patterns for examining clouds typically resembled a series of stacked SLRs below-cloud, within cloud, above cloud, and within aerosol. Typically, the patterns were used together with a series of saw-tooths through the cloud to further characterise the variability of the cloud top and cloud base and to provide detailed characterisation of cloud microphysical parameters within cloud and at cloud-top from in-

situ measurements (effective radius, LWP, LWC).

SLRs just below cloud base and just above cloud base were used to investigate CCN budgets, closure and aerosol loss due to scavenging. SLRs below-cloud, in-cloud and above-cloud measured CCN, cloud droplet size distributions and drizzle size distribution below cloud base



to provide information on the entrainment process, the influence of entrainment on cloud microphysics and constraints on BBA entrainment rates into cloud top. Vertical profile/saw-tooth/stepped profile measurements were made of the size distribution of cloud droplets and

precipitation over the diameter size range 2 µm to 6 mm capturing cloud droplets and precipitation. The onboard AMS/SP2/OPCs were switched between the CVI inlet to measure droplet residuals and the total inlet to determine the size and composition of the nucleation scavenged and interstitial aerosol as a function of position and height in the cloud. Measurements higher in the cloud together with turbulence measurements examined the evolution of the cloud microphysics as condensation growth and coalescence occur. Precipitation susceptibility ($-d(\ln P)/d\ln(N_{CCN})$, in which P is the precipitation rate and $N_{CCN}$ is the CCN number concentration) can be determined from

measurements of precipitation rate, cloud water contents, cloud thickness and CCN concentrations. By compositing cloudy columns with a given thickness (or LWP), the relation between changes in precipitation and aerosol perturbations could be made.

### 4.3 Planning logistics

All satellite overpasses, satellite observations from previous overpasses, model data, and observations from the AMF and from the KNMI lidar installed at the airfield were available to the planning teams (see Sect. 2 and 3). Owing to the high intensity of the flying programme,

the flight planning teams were separated into two; an aircraft-based team flying the mission and a second ground-based team which prepared flight plans for the forthcoming flight. The team on the ground also was responsible for sending updates to the aircraft via satcom providing updates of the cloud conditions from the geostationary satellites and measurements from the surface-based instrumentation. After a de-brief of the flight, the ground-based team and aircraft-based teams then swapped roles so that each team "owned" the flight from inception, through planning and execution. As per standard campaign operating procedures, a running tally of hours allocated to specific aerosol

characterisation, cloud characterisation, aerosol-radiation and aerosol-cloud interactions and other aspects such as POC investigations was maintained during the campaign so that future flights could target any science gaps in the key objectives.

### 5 Summary of the Flights performed

Twenty-eight science flights were performed on 18 days during CLARIFY-2017 for a total of around 99 hours (Table 2); the geographic distribution of the flight tracks for CLARIFY-2017 is shown in Figure 7.

***Insert Table 2***
***Insert Figure 7***

While the climatological mean of the AOD shows a maximum almost directly east of Ascension Island (Figure 1), in practice Figure 7

shows that the flights were performed in various directions because of the filament-like nature of the aerosol plume on any specific day. The aerosol optical depth measured at Ascension Island via the Cimel and microtops sunphotometers at 500 nm ranged from values of 0.14 to 0.54 (Table 2, Figure 8) and a t-test value of 0.9879 indicates that the AOD was not significantly different from the long term data from the AERONET Cimel.

***Insert Figure 8***

A more detailed summary of each of the flights in terms of the meteorological conditions, aerosol vertical profiles and manoeuvres is available in the Supplementary Information.



## 6 Key results

With reference to the Key Objectives of Section 1.4, the following sections report the key results from our analyses.

### 6.1 Vertical profiles

CLARIFY-2017 was able to show that the vertical structure is quite complex with aerosols existing either solely in the MBL, solely in the residual CBL or existing in both the MBL and residual CBL (Figure 9).

*** Insert Fig 9***

Figure 9 shows that the CLUMP model is generally able to represent the distribution of aerosol in the MBL and the residual CBL. Figure 9 shows that the one notable exception is when a pocket of open cells (POC) was observed over Ascension Island. During the POC event towards the end of the measurement campaign, the model does not accurately represent the close-to-pristine nature of the MBL (see Abel

*et al.*, 2020).

A hierarchical cluster analysis was performed based on the mean and maximum BBA concentrations and the altitude of the maximum concentrations in the free troposphere and the mean concentration within the boundary layer for each flight. The cluster analysis based on these criteria revealed two distinct groups, with the first group (G1) including flights C028-C032 (16-19 August, 2017) corresponding to the period when the aerosol was solely in the MBL with a mean concentration over the size range $0.1 - 3.0$ μm (Table 2) measured by the

PCASP instrument on the FAAM aircraft of 685 cm$^{-3}$ in the MBL but just 35 cm$^{-3}$ in the free troposphere. The second group contained two sub-groups with flights C034, C035, C042 and C047-C050 i.e. those showing little aerosol in the MBL (~78 cm$^{-3}$), but much in the residual CBL (~884 cm$^{-3}$). These are denoted group G2. The mean synoptic geopotential height based on these two clusters is shown in Figure 10.

***Insert Figure 10***

Figure 10 shows that, although southeast winds associated with a subtropical high dominated in the MBL at the location of Ascension Island

(as indicated by the 925 hPa geopotential height contours) for both groups G1 and G2, the locations of the high pressure centres were different. In G1, the centre of the high pressure was located around 40° S, $0 - 20$° E while under G2, the centre of the high pressure was around 30° S, $0 - 10$° W. For both G1 and G2, the MBL around Ascension Island can be influenced by air of continental origin, but the MBL in G2 is also influenced by air recirculating around the sub-tropical high that does not pass over the African continent owing to the non-geostrophic and divergent flow around high pressures. This recirculation characteristic of G2 appears to explain the relatively clean

MBL during the periods 21-25 August and 31 August – 4 September (Figure 9). The geopotential height fields at 700 hPa show significant differences in airflows between the two clusters. In G1, where the cases were represented by a relatively clean free troposphere, there was no clear high pressure to the southeast of Ascension Island. In contrast, in G2, high pressure extended from Namibia/Angola to the island with associated strong easterly winds that transports smoke from the African continent to the Ascension Island region in the residual CBL (Table 2).

This analysis is enhanced by an analysis of back-trajectories (HYSPLIT4, using ERA5 reanalyses (Hersbach, 2016)) presented in Figure 11, which shows trajectories initiated every 1-hour for August 2017 at 50m (lower MBL), 330m (mid-MBL), 1000m (upper-MBL) and 2000m (lower residual CBL).

***Insert Fig 11***




Back-trajectories originating at 50 m and 330 m over Ascension Island indicate that the flow at these levels originates in oceanic regions. At 1000 m (upper-MBL), the back-trajectories indicate some influence from land areas in northern Namibia and Angola; these areas experience seasonal burning, (e.g. Abel *et al.*, 2003) and thus some of the BBA detected in the MBL is likely to come from these regions.

At 2000 m, in the residual CBL, the back-trajectories indicate an easterly flow and hence more northerly source of BBA (northern Angola, Gabon, Congo, Democratic Republic of Congo, Equatorial Guinea and Cameroon) from an area where fires are most prevalent. BBA in in the residual CBL only influences the microphysics of low-lying clouds after those aerosols are entrained into the MBL. Thus, any inferences of aerosol-cloud-interactions that depend on relationships between the column AOD and CDNC (e.g. Quaas *et al.*, 2008) may be erroneous in this region (Stier, 2016).

As demonstrated in Figure 9, comparison of the shape of the mean vertical distribution of aerosol extinction derived from the aircraft-based

EXSCALABAR measurements and CLUMP NWP model shows a reasonable agreement, although generally the model does not extend the vertical distribution of aerosol high enough and there is rather too much aerosol in the MBL. Simulations with the atmosphere only (i.e. not coupled to the ocean model) version of HadGEM3 (Hewitt *et al.*, 2011) indicate that the discrepancy in the vertical distribution of aerosol is very likely due to the lack of account of aerosol radiative effects, in particular the model's neglect of aerosol absorption that "self-lofts" the air containing the BBA (Figure 12).


*** Insert Figure 12 ***

While this 'self-lofting' has been recognised for decades in smoke plumes (e.g. Westphal and Toon, 1991), the near-continental large-scale nature of the ascent rate and the counterbalancing descent elsewhere hints at a further impact of aerosol beyond aerosol, direct, indirect and

semi-direct effects; that of teleconnections. Because atmospheric dynamics are constrained by physical laws of conservation of energy and momentum, any large-scale lifting of air-masses must be balanced by large-scale descent of air-masses elsewhere. Figure 12b shows the spatial extent of the ascent (or the reduction in subsidence) over the region caused by absorption of the BBA. This suggests that the lack of inclusion of the radiative impacts of absorbing aerosols owing to computational constraints may have consequences on the performance of NWP models in accurately representing mean vertical velocities. In turn this may influence the strength of the Hadley and Walker

circulations. However, it is acknowledged that the atmosphere-only simulations shown here neglect any dynamical changes that may be induced through changes in surface land temperatures or sea surface temperatures (SST) which can induce changes to the thermally direct atmospheric circulation (e.g. Roekner *et al.*, 2006, Sakaeda et al. 2011) and ocean heat transport which has a large impact on the overall dynamical response (e.g. Hawcroft *et al.*, 2018). The impacts of aerosol direct and semi-direct effects are also investigated in regional high-resolution regional climate models under the AEROCLO-sA measurement campaign (Mallet *et al.*, 2020).

The performance of the HadGEM3 NWP model with CLUMP aerosol scheme compares favourably with the other modelling tools used in forecasting the aerosol spatial distribution. Note that, under the ORACLES project, Shinozuka *et al.*, (2020) performed a multi-model analysis of vertical profiles of BBA against observations from ORACLES during the 2016 deployment when the NASA P3 aircraft was operating from Namibia over a wide area of the SE Atlantic closer to the African continent.. Their results suggest that, for that region, each of the models analysed presents its own strengths, weaknesses and biases but one common feature is that all models tend to underestimate

the height of the base of the smoke layer. This does not appear to be the case with the CLUMP simulations for the CLARIFY-2017 region, where the bottom of the residual CBL aerosol layer frequently corresponds to the top of the MBL (Figure 9), forcing an accurate lower boundary for the residual CBL plume. The results comparing the CLUMP model to the observations suggest reasonable agreement in aerosol peak concentrations and in the total integrated extinction (Figure 9) but, as noted earlier, the neglect of aerosol absorption in the NWP model appears to result in a peak aerosol concentration and upper bound of the plume that is approximately 1-2 km too low.





At the top of the MBL, a strong temperature inversion provides a strong energetic barrier to vertical mixing (e.g. Wood and Bretherton, 2006) providing an effective cap to cloud vertical extent. Figure 13 shows the observed boundary layer height as diagnosed from radiosondes launched from Ascension Island and the modelled cloud liquid water path. The Met Office NWP model represents the boundary layer height adequately over Ascension Island. The boundary layer height is important in retrievals of above cloud aerosol properties from SEVIRI as errors impact the amount of water vapour above cloud that is assumed in the retrieval algorithm (Peers *et al*., 2019). Figure 13 also shows

the cloud droplet number concentration as measured by the CDP instrument over the field campaign. There is clear evidence of the influence of aerosol-cloud-interactions in the cloud-droplet number concentration. The cloud droplet number concentration is at its highest (~mean of 360 cm$^{-3}$) at the start of the measurement period, when the BBA is present in large quantities in the MBL, but abruptly transitions to its lowest value (~15 cm$^{-3}$) when the MBL is close to pristine. Low CDNC values are also found in the measurements when the POC is present towards the end of the deployment period (Fig 13).


***Insert Fig 13***

        As a result of the strong vertical shear in wind-speed, the time for an air-parcel from leaving the African continent to reaching Ascension Island is shorter for aerosol higher up in the residual CBL than lower down in the CBL or in the MBL. Thus, in general, the aerosol at lower

altitudes can be significantly older compared to that located at elevated altitudes (see analysis of ORACLES data by Dobracki *et al*; 2020). There is clear evidence from both Wu *et al*. (2020a) and Dobracki *et al*. (2020), that aerosol higher up in the residual CBL exhibits a higher SSA (i.e. it is less absorbing on a per particle basis) than that lower down. Taylor *et al* (2020) also show that the mass absorption coefficient in the CLARIFY domain does not vary significantly with altitude. Wu et al (2020a) and Taylor et al. (2020) propose that the partitioning of a higher fraction of inorganic ammonium nitrate onto the existing particles at the colder temperatures associated with the higher altitudes

explains the vertical structure in SSA in the region above Ascension Island. This explanation differs from that of Dobracki et al (2020) who suggest loss of scattering organic material from the BBA as the aerosol ages based on data from the farther-ranging the ORACLES flights which encompassed a wider range of aerosol ages (see also section 6.5).

**6.2 Analysis of aerosol size distributions**

        Prior to the ORACLES, AEROCLO-sA, LASIC and CLARIFY-2017 campaigns, airborne measurements of BBA size distributions in the

region were sparse. Haywood *et al*. (2003) documented the size distribution of BBA during SAFARI-2000 both close to emission source, off the coast of Namibia and in the vicinity of Ascension Island; both of these cases are of relevance for CLARIFY-2017. Although the data presented here is quality assured, the specific analyses performed during CLARIFY-2017 inevitably differ due to different sampling locations, sampling periods, different case studies etc. Therefore, we present a composite of the models that are used to fit data in these studies to allow a quantification of the error introduced by the assumptions used in each analysis. Generally, these models use log-normal

fits of the form:-

$$\frac{dn_i(r)}{d\ln r} = \frac{n_i}{\sqrt{(2\pi)}\ln\sigma} \exp\left[-\frac{(\ln r_i - \ln r_n)^2}{2(\ln\sigma)^2}\right] \tag{1}$$

        In which $n_i(r)$ represents the number of aerosols of radius $r$ for mode $i$, $r_n$ represents the geometric mean radius and $\sigma$ is the geometric

standard deviation. Table 3 shows examples of the fits of this equation to the measured or retrieved size distributions during SAFARI-2000 and CLARIFY-2017, effective refractive indices and the resultant single scattering albedo.

***Insert Table 3***





The size distributions for BBA from Peers et al. (2019) and Wu et al. (2020) are consistent with those determined from SAFARI-2000 although Haywood et al., (2003) chose to describe the accumulation mode with two log-normal distributions, rather than a single log-normal distribution. The corresponding refractive indices retrieved over the CLARIFY period (16 August – 7 September 2017) derived from AERONET Version 2 algorithms for the Ascension Island site are 1.47-0.020i at a wavelength of 550 nm. The smaller value of the real and imaginary part of the refractive indices from AERONET compared to the in-situ retrievals documented in Table 3 likely reflect that they

represent column averaged properties and hence there is a contribution from aerosol components such as sea-salt and sulphate of DMS origin within the MBL (Wu *et al*., 2020a; Taylor *et al*., 2020). Wu *et al*. (2020a) state a mean SSA at 550 nm of approximately 0.81 at 2 km altitude in the residual CBL rising to 0.86 at 5 km altitude and assign the difference in SSA to the thermodynamic impact of temperature on the partitioning of inorganic nitrate into the aerosol phase.

### 6.3 Analysis of aerosol chemical properties

The composition of the aerosol in different layers was measured with an Aerodyne Compact Time-of-Flight airborne AMS (C-ToF AMS, Table 1), which provided organic mass, nitrate, sulfate and ammonium mass concentrations, and an SP2, to determine the BC mass concentration. Vertical profiles of these different chemical components averaged across each of the regimes, together with campaign-average compositions, are shown in Figure 14. These data have enabled a detailed characterisation of the composition of aerosol in the region (Wu *et al*., 2020a). During periods when the residual CBL is filled with BBA, large quantities of predominately organic aerosol are

present. The composition fractions (average ± standard deviation) were OA(61 ± 5) %, BC(13 ± 3) %, $SO_4$(11 ± 4) %, $NO_3$(8 ± 3) % and $NH_4$(7 ± 2) % (Wu *et al*., 2020a; Taylor *et al*., 2020) suggesting a BC/OM ratio of around 0.21. Notably, the inorganic components are present in significant mass concentrations and the fraction of ammonium and nitrate present increases with altitude.  The MBL displays a greater proportion of sulfate than the residual CBL while the mass fractions of nitrate, BC and OA are lower in the MBL. This increased sulfate mass fraction in the MBL is likely due to the formation of sulfate from DMS oxidation in the MBL since similar concentrations are

present during clean (MBL) periods. The main chemical properties and processes governing the particulate chemistry are discussed in detail by Wu *et al*., (2020a).

***Insert Fig 14***

### 6.4 Analysis of in-situ aerosol optical properties

A key objective of CLARIFY-2017 was to assess, to the highest degree of accuracy possible, aerosol optical properties, with a particular focus on the aerosol SSA owing to its strong influence on the aerosol direct radiative effect. Such an objective precludes the use of filter-based observations which are subject to a wide range of empirical corrections (e.g. Bond *et al*., 1999, Davies *et al*., 2019) and have been shown under some conditions to yield uncertainties in absorption of over 200% (Lack *et al*., 2008, Cappa *et al*., 2008). Recognising the

crucial importance of aerosol absorption for understanding aerosol-climate interactions, the UK Met Office has developed and tested a new state-of-the-art spectroscopic instrument for accurate measurement of aerosol optical properties. The instrument (EXSCALABAR, Table 1), employs multiple Cavity Ring-Down extinction and Photoacoustic Absorption Spectrometers to determine multi-wavelength measurements of optical attenuation coefficients for dry, humidified and thermally-denuded aerosols to high precision and accuracy (Davies *et al*., 2018; 2019; Cotterell *et al*., 2019; 2020).  Davies *et al*. (2019) examined the biases in filter-based retrievals of the aerosol absorption.

While these biases were far more modest than those derived by Lack *et al*. (2008) and Cappa *et al*. (2008), they remained of a significant level (~20%) for aged BBA and depended on the correction scheme; the biases were reduced to levels of <11% using advanced two-stream





radiative transfer correction schemes (Mueller et al., 2014), but took values up to 21% when using the more common correction scheme of Bond *et al*. (1999). For other aerosol sources such as urban aerosols, Davies *et al*. (2019) report an overestimation of absorption from filter-based measurements using the correction scheme of Bond *et al*. (1999) of up to 45%.


Davies *et al*. (2019) performed an analysis of the SSA of aerosol dominated by BBA in both the MBL and the residual CBL derived from EXSCALABAR and presented detailed probability distributions of the derived SSA, finding mean values of 0.84, 0.83 and 0.81 at 467, 528, and 652 nm respectively. Wu *et al*. (2020a) extended this analysis, reporting column weighted dry SSAs derived from EXSCALABAR and find a mean and standard deviation of $0.85 \pm 0.02$, $0.83 \pm 0.03$ and $0.82 \pm 0.03$ at 405, 550 and 658 nm respectively in the residual CBL,

with evidence that the SSA increased with altitude in the residual CBL. Interestingly, these mean values are in agreement with those from radiometric measurements, which do not rely on filter-based absorption instrumentation, derived from nine above-cloud flights of the NASA P3 aircraft during the 2016 and 2017 ORACLES campaign (Cochrane et al., 2020).

In the MBL, Wu *et al* (2020a) report SSAs and standard deviations derived from EXSCALABAR of $0.86 \pm 0.02$, $0.85 \pm 0.03$ and $0.84 \pm$

0.03 at 405, 550 and 658 nm respectively. Zuidema *et al*. (2018b) report SSAs from the ARM site (at ~330 m ASL i.e. residing in the MBL) of $0.78 \pm 0.02$ for August 2016-2017 and $0.81 \pm 0.03$ for September 2016-2017 (interquartile range) at 529 nm, which suggest stronger absorption than the study of Wu et al., (2020). Zuidema *et al*. (2018b) acknowledge that the filter-based systems are dependent on the artefact-correction algorithm and use the mean of the correction from Virkkula (2010) and Ogren *et al*. (2010) algorithms. However, their filter-based measurements agreed with measurements made with those from an AERODYNE CAPS-SSA instrument deployed in July-

September 2017; both yielded values of 0.77 at 529 nm. Without these additional measurements, the apparent discrepancy between the ARM and CLARIFY measurements could have be attributed to the remaining biases associated with filter-based correction algorithms. Davies *et al* (2019) showed that these correction algorithms typically overestimate aerosol absorption and without moving to more advanced 2-stream correction algorithms (Müller *et al*., 2014), these correction algorithms underestimate the SSA by around 0.03 to 0.04 at 550nm for measurements made during CLARIFY-2017. In addition, the MBL mass absorption coefficients are consistent between CLARIFY

(Taylor et al., 2020) and LASIC (Zuidema et al., 2018b), indicating that it is the scattering measurements that differ between the two campaigns, rather than the more challenging absorption measurements. Work is currently underway to fully investigate these discrepancies. One possibility is that the impactor used in sampling the aerosol for CLARIFY may not correspond exactly to that for the AMF inlet (1.3 and 1.0 µm aerodynamic diameter respectively). Thus, a fraction of super-micron sea-salt aerosols may increase the SSA for the CLARIFY-2017 measurements.


It is clear from the results of Wu *et al*. (2020a) and Taylor *et al* (2020) that the many of the aged BBA particles in the vicinity of Ascension Island consists of a core of black carbon with a thick coating of organic and inorganic material (shell-core diameter ratio ranging from around 2.3 at the surface, to approximately 2.6 at 5 km ASL). Over the wavelength range 405 – 660 nm there is little evidence of absorption by organic 'brown' carbon, but there is clear evidence of absorption enhancement via a lensing effect whereby incident radiation is focussed

onto the absorbing core of black carbon; this effect was also suggested by Zuidema *et al* (2018b). Taylor *et al* (2020) show that aerosol optical properties are not well represented when using the volume weighting of refractive indices that is currently used in many GCMs. While the models documented in Table 3 utilise volume weighting of refractive indices, the resultant mass absorption coefficient (i.e. the mass-normalised aerosol absorption cross section) using a straightforward Mie theory model with these volume-weighted effective refractive indices does not agree with measurements derived from the EXSCALABAR, AMS and SP2 instruments (Taylor *et al*., 2020).

Internally consistent optical closure of both the optical parameters and the mass absorption coefficient can be improved using core-shell





Mie scattering treatment of a black carbon core and an organic/inorganic coating but can be most accurately reconciled using more complex semi-empirical parameterisations of mixing state (Taylor *et al.*, 2020).

### 6.5 Aerosol ageing

BBA measured in the vicinity of Ascension Island was always very aged (>7 days from emission; Wu *et al.*, 2020a) and consisted of a thick
coating of organics/inorganics surrounding an insoluble black carbon core (Taylor *et al.*, 2020). While these measurements alone do not allow us to estimate the impacts of ageing on aerosol physical and optical properties, the same instrumentation has been flown during other campaigns e.g. Methane Observations and Yearly Assessments (MOYA; Allen *et al.*, 2017; Davies *et al.*, 2019; Wu *et al.*, 2020b) that made measurements much closer to the source regions of the biomass burning over continental Africa (Davies *et al.*, 2019, Wu *et al.*, 2020b). Wu *et al.* (2020b) use identical measurement systems to those used during CLARIFY-2017 and report mean shell/core diameter ratios for BBA
of as little as 1.07 (stdev 0.10) for BBA less than 30 minutes subsequent to emission, increasing to 1.39 (stdev 0.06) for BBA 3-6 hours subsequent to emission and 1.66 (stdev 0.07) for BBA 9-12 hours subsequent to emission. The shell/core ratios of 2.3-2.6 determined during CLARIFY-2017 suggest that the coating has continued to thicken as the BBA ages and the constituent components become increasingly internally mixed. Davies *et al.* (2019) also use nominally identical EXSCALABAR instrumentation during MOYA to determine a BBA SSA of around 0.91 at wavelengths close to 550 nm for BBA that has aged by 9 - 12 hours since emission. Scanning electron microscope
measurements made as long ago as SAFARI-2000 suggested that, on emission, black carbon consisted of individual spherules in chain-like structures (Posfai *et al.*, 2003). Owing to surface tension effects, these chain structures collapse to more compact cores when coated by organic aerosol that was either formed at source or through the condensation of semi-volatile organic species within a few seconds from emission (e.g. Posfai *et al.*, 2003; Abel *et al.*, 2003). The black-carbon chainlike structures have a higher fractal dimension and a higher absorption efficiency compared to the more compact cores (Chakrabarty and Heinson, 2018). Together with the condensation of organic or
volatile inorganics, which are predominantly scattering in nature, one might expect the SSA to increase with time (e.g. Abel *et al.*, 2003). However, this condensation of scattering species can have the opposite effect, acting effectively as a lens focussing radiation on the absorbing core. Additionally, oxidation and nitration of the organic components could lead to an increase in absorption by 'brown' carbon (e.g. Saleh *et al.*, 2015), but conversely photochemical bleaching of BBA particles has been noted in laboratory studies (e.g. Zhong and Jang, 2014).

Trajectory simulations prove that aerosol high up in the atmosphere is generally younger (Dobracki et al., 2020; section 6.1). From ORACLES measurements, Dobracki et al., (2020) suggest that the changes in the aerosol SSA as the aerosol ages is due to a reduction of organic material through evaporation. However, the vertical profile of the chemical composition of the BBA may be complicated by differences in the thermodynamic structure of the residual CBL and the condensation of inorganic nitrate into the aerosol phase (Wu et al., 
2020a; Taylor et al., 2020). Taken together, at the process level, the competing effects of fractal chain collapse, the evolving lensing effect from increasing coating thicknesses, the changing absorption of the BrC coating, and the details of condensation/evaporation of volatile aerosol components make the aging process particularly complex and not attributable to a single change in aerosol microphysics. What is clear, is that the BBA measured during CLARIFY appears to be more strongly absorbing than that emitted at source, indicating that as BBA ages, the mechanisms that increase absorption outweigh those that may decrease absorption. However, at the time of writing, the contribution
of the processes documented above remains an open issue and will undoubtedly be the subject of further work.

### 6.6 Aerosol-radiation interactions

Aerosol-radiation interactions have been investigated at several scales. Peers *et al.* (2019, 2020), has developed a novel above cloud aerosol detection algorithm from the geostationary SEVIRI instrument. Importantly, this retrieval accounts for the impacts of water vapour in the





relatively wide SEVIRI spectral bands by assimilating humidity profiles from the Met Office NWP global model leading to improvements
in the accuracy of the retrievals (Chang and Christopher, 2016). A comparison against above-cloud retrieval algorithms developed from MODIS (Meyer *et al.*, 2015) has been performed revealing some systematic differences, but overall the agreement in cloud and aerosol properties is satisfactory (Peers *et al.*, 2020). The geostationary nature of the SEVIRI satellite instrument means that, unlike polar orbiting satellite retrievals which require precise colocation, coherent comparisons between aircraft and SEVIRI retrievals are possible. A number of cases have been investigated, with encouraging agreement between aircraft and SEVIRI retrievals (Peers *et al.*, 2020). Additional work
has confirmed the strong magnitude of the above cloud direct radiative effect using OMI and MODIS (De Graaf *et al.*, 2019) and a comparison of the above cloud direct radiative effect from various space-borne sensors has been performed (De Graaf *et al*, 2020).

Herbert *et al.* (2020) examined semi-direct effects (sometimes referred to as rapid adjustments to aerosol-radiation interactions) from absorbing aerosol layers overlying the southern Atlantic stratocumulus deck using large eddy simulation (LES) modelling. Herbert *et al.* (2020) diagnose SDEs (W m$^{-2}$) from changes in the cloud resulting in modelled fast-feedbacks. SDEs diagnosed in this way in the region
appear to have a strong diurnal cycle, peaking in the morning, so daily-averaged SDE is much weaker than instantaneous values would suggest. Aerosol layers located immediately above the cloud exert strong SDE by affecting the temperature inversion at the top of the MBL and reducing the entrainment of air into the stratocumulus. The LES simulations suggest that this SDE weakens considerably with increasing distance between aerosols and clouds, with SDE exerted by aerosol layers 250 m away from the cloud top roughly half of that of layers located just above the cloud top, and almost no SDE when the aerosol layer is 500 m above the cloud top. An analysis of lidar profiles from the NASA Cloud–Aerosol Transport System (CATS) lidar
(5 km resolution, V3-00, mode 7.2, level 2 Daytime Operational Layer Data Product, 1064 nm wavelength) suggests that in some 27% of cases the aerosol base is within 500m of the cloud top, and therefore close enough to exert a SDE (Figure 15). Additionally, in 22% of cases the whole BBA layer is within 2 km of the cloud top, yet only 3% of cases are within 1 km. Of course, this analysis is over a far greater area than that sampled in the vicinity of Ascension Island (Figure 7) and includes areas off the coast of Angola and Namibia where "clear slots" (Hobbs, 2003; Haywood *et al.*, 2003; Redemann *et al.*, 2020) are more evident.


***Figure 15***

Note that these results contrast with others. The global model focussed work of Che *et al.* (2020a), suggests that aerosol SDEs are stronger than those from direct and indirect effects (see section 6.9). In common with earlier studies (e.g. Johnson *et al.*, 2004), the magnitude and
the sign of the SDE are dependent on the relative location of BBA and clouds, as BBA can either increase the underlying cloud LWP or decrease the surrounding droplet numbers depending on whether the BBA are above or inside the cloud. Zhang and Zuidema (2019) also found that the cloud-top inversion was often weaker when aerosol was likely present in the free-troposphere, during August 2016-2017, rather than stronger, and attributed this to a meteorological effect; the residual of the CBL containing the aerosol is also often cooler than the air it is replacing. Such disagreements may stem from the relative ability of different models to resolve changes in boundary layer
moisture and in the temperature inversion above stratocumulus, the lack of weakening of subsidence from BBA heating in LES studies (Myers and Norris 2013), and confounding effects from meteorology.

### 6.7 Aerosol-cloud interactions

The magnitude of aerosol-cloud-interactions has been the subject of intense debate over the past two decades. While there is clear evidence of the Twomey effect (Twomey, 1977) from recent comprehensive satellite assessments of ship-tracks and degassing volcanoes (e.g.
Malavelle *et al.*, 2017; Christensen *et al.*, 2014, Toll *et al.*, 2017; 2019) there is generally a lack of clear evidence of the "cloud lifetime" or Albrecht effect (Albrecht, 1989) as the impact appears to be strongly dependent on the atmospheric state (e.g. Chen *et al.*, 2015; Toll *et al.*, 2019). On a cloud-by-cloud basis, there is considerable evidence from observational studies (e.g. ***refs***) that if there is an increase in





the CDNC at a fixed cloud liquid water content, then the auto-conversion process by which cloud droplets grow to a size big enough to initiate precipitation should be inhibited .


***Insert Figure 16***

On a statistical basis throughout the CLARIFY-2017 campaign, a strong cloud response to aerosol loading is observed (Figure 16; see Barrett et al., 2020b for more details), where increased aerosol concentrations ($N_a$) under polluted MBL conditions ($N_a = 442 \pm 525$ cm$^{-3}$)

resulted in greater cloud droplet concentrations (CDNC = $122 \pm 86$ cm$^{-3}$) than was observed for clean conditions ($N_a = 79 \pm 96$ cm$^{-3}$, CDNC = $26 \pm 53$ cm$^{-3}$), where clean and polluted conditions are defined using a CO threshold of 83 ppb. There was also a corresponding influence on the cloud droplet effective diameter, consistent with the Twomey effect (Twomey, 1977), where the cloud droplets were observed to be significantly smaller under polluted conditions than compared to clean conditions (19 µm and 36 µm respective median values) despite the observation from the LASIC AMF that the LWP generally increases with pollution levels for CLARIFY time period (Barrett et al., 2020b).

Generally the cloud top altitudes (Zct) were found to be higher under polluted conditions than clean conditions. The influence of pollution on precipitation is less clear, however, approximately 42% of in cloud data points (defined as total water content from the Nevzerov probe > 0.01 g m$^{-3}$) contained drizzle (defined where drizzle water content > 0.01 g m$^{-3}$) in clean conditions compared to approximately 24% in polluted conditions. Care must be taken when interpreting these initial results as they may be influenced by the overall sampling strategy and the potential influence of the BBA impacts on reduced subsidence (Myers and Norris, 2013); further work is ongoing examining the

co-variability of cloud microphysical and meteorological influences. However, these conclusions do appear consistent with data derived from independent carbon monoxide (CO) and precipitation data obtained from the ARM mobile site and precipitation measurements from the Met Office located on Ascension Island. We extend the analysis period to August-September for 2016 and 2017 and precipitation data from the Met Office at Ascension Island which is less likely to be impacted by orographic effects than precipitation at the LASIC AMF site. Figure 17 shows the resulting distributions of CO concentration for precipitating and non-precipitating days.


****Insert Figure 17***

Figure 17 suggests that precipitation does appear to be inhibited in polluted days when compared to non-polluted days as evident from the means of the distributions (77.8±30.4 (2 stdevs) for the 51/122 precipitating days and 95.7±60.8 (2 stdevs) for the 71/122 non-precipitating

days). The mode CO is identical for the precipitating and non-precipitating days while the CO distribution for the non-precipitating days shows evidence of a far longer tail to high values of CO. Note also that the patchy nature of precipitation over the Ascension Island region means that non-precipitating days will likely contain days when there is precipitation in the vicinity, but there is no precipitation detected at the Met Office site. Of course, we acknowledge that this is a purely statistical analysis that uses CO as a proxy for CCN and does not account for cause and effect, nor does it take account of covariability that might influence precipitation such as the boundary layer height,

cloud top height and LWP but these results appear consistent with the results from Zhang and Zuidema (2019), who analysed disdrometer data and polluted/non-polluted conditions determined from the AMF LASIC deployment. The ability of BBA to act as CCN over the SE Atlantic is further elucidated under the ORACLES project (Kacarab *et al*., 2020).

Certain of the flight days provided a particular wealth of information on cloud properties in the region. On 19 August 2017, for example, a

large cloud feature south of Ascension Island about 1.5° in size was sampled at five different altitudes during straight-and-level runs. The aircraft followed a line of strong echoes on its flight-instrument radar. Droplet size distributions and out-of-cloud samples of the aerosol size distributions were obtained at each level up to and above the cloud top height which was around 2.5km in altitude. This case study as



the cloud changed from an overcast stratocumulus system to organised convective clouds is documented by Cui *et al.* (2020) and a model-observation comparison by Gordon *et al.* (2020).

### 6.8 POCs

The role that the free-tropospheric BBA plumes observed over the SE Atlantic play in modulating the evolution of the underlying clouds via microphysical perturbations, is dependent on where and when the BBA plumes mix down into the boundary layer. CLARIFY airborne and LASIC ground based measurements from a case-study of both a Pocket of Open Cells (POC) and the surrounding stratiform cloud topped boundary layer, highlighted that the efficiency of this entrainment of aerosol can depend on the form of the underlying cloud structure (closed vs open cellular convection), with a marked reduction in entrainment of BBA in the region of open cells (Abel *et al.*, 2020). An analysis of satellite imagery in Abel *et al.* (2020) demonstrates that these open cellular cloud regions occur regularly in the offshore environment surrounding Ascension Island during September. If the findings from this case apply more broadly, then this low-susceptibility of open cells to intrusions of overlaying BBA could have important implications for aerosol indirect effects in the region, especially given that global climate models are generally not capable of simulating mesoscale features such as POCs, due to their coarse resolution and often relatively simplistic representation of aerosol-cloud-precipitation interactions.

### 6.9 Large-scale model-focussed investigations

The data collected during the CLARIFY campaign are already proving a valuable resource for modelling studies. In addition to the modelling work in support of flight planning (section 1), preparatory work funded by the project involved testing the GLOMAP-mode aerosol microphysics scheme in the Unified Model at convection-permitting resolution over the south-east Atlantic (Gordon *et al*, 2018). The scheme was evaluated against satellites and data from the ARM site at Ascension Island and shown to perform well. The Unified Model global configuration was subsequently also shown to predict properties of smoke aerosol realistically in an evaluation against CLARIFY measurements of extinction (Che *et al*, 2020a) and ORACLES 2016 measurements (Shinozuka *et al*, 2020).

One aspect that is interesting is that the direct radiative effect of above cloud aerosol appears to be fairly independent of model resolution. This might be thought of as somewhat of a surprise because finer resolution models can include higher grid-box mean AODs and CODs and they might therefore be expected to give a wider range of direct radiative effects when compared to coarser resolution versions of the model (Figure 18).

***Figure 18***

The simulations shown in Figure 18 are nudged simulations with the Unified Model performed for August-September 2006 to coincide with POLDER observations. The coarser resolution models show a more spatially homogeneous spatial distribution of the DRE than the higher resolution models, but when averaged over the domain, the probability distribution of the direct radiative effect is a similar shape. That the model direct effects do not diverge as a function of model resolution is a testament to the validity of utilising a model with identical underlying physics in the Unified Model framework.

A complementary global modelling paper (Che *et al*., 2020a) examines a wider area of the south-east Atlantic in an atmospheric UK Earth System Model (UKESM1) configuration of the Unified Model, at N96 resolution, representative for typical climate model simulations to assess regional and global climate impacts over a longer time-period. This work highlights the complex interaction of radiation, microphysics and dynamical feedbacks. Decomposition of the radiative effects shows that the regional direct radiative effect is generally positive when the biomass burning plume is above the stratocumulus deck (with July-August average +7.5 W m$^{-2}$), as the surface albedo of





the underlying clouds is fairly high (e.g. Keil and Haywood, 2003). However, in UKESM1, rapid adjustments (semi-direct effect) enhance cloud albedo and more than compensate the direct effects, resulting in a net negative cooling effect over the region (July-August average -0.9 W m$^{-2}$). Microphysical effects of aerosol-cloud further increase cloud albedo and associated negative radiative effect. In the global mean, rapid-adjustments due to biomass burning (semi-direct effects) appear negligible. In separate work based using the same model

configuration, Che *et al.* (2020b) perform a source attribution of CCN and resulting cloud droplet numbers for the CLARIFY domain. Che *et al.* (2020b) estimate that during biomass burning season, upper tropospheric binary nucleation between sulphuric acid and water (Vehkamäki *et al.*, 2002) and BBA contribute a similar amount of CDNC and are the most importance two sources of CCN in this region. This highlights the importance of upper tropospheric nucleation and subsequent subsidence in subtropical areas for the local cloud regime (e.g. Clarke, 1993) and highlights a route for significant microphysical aerosol effects on clouds, at considerable distance from

anthropogenic source regions.

In separate work based using the same model configuration Che *et al* (2020b) perform a source attribution of CCN and resulting cloud droplet numbers for the CLARIFY domain. Che *et al*. (2020b) note that CCN from upper tropospheric binary nucleation contributes approximately 50% to BBA impact on droplet numbers in the region, with negligible contributions from sea salt and boundary layer nucleation. This highlights the importance of upper tropospheric nucleation and subsequent subsidence in subtropical areas for the local

cloud regime (e.g. Clarke, 1993) and highlights a route for significant microphysical aerosol effects on clouds, at considerable distance from anthropogenic source regions.

Further work on regional modelling includes the coupling of the GLOMAP-MODE two-moment aerosol microphysics to the CASIM two-moment cloud microphysics scheme, which allows more refined studies of the indirect effect (Gordon *et al.*, 2020). The current intensity of modelling activity suggests that the CLARIFY dataset will be a valuable resource for model evaluation for many years to come.

The measurements obtained during the CLARIFY measurement campaign will also, together with data from ORACLES, LASIC and AEROCLO-sA, contribute a key constraint on the representation of biomass burning aerosols in current climate models as part of the ongoing AeroCom Aircraft intercomparison study (https://wiki.met.no/aerocom/phase3-experiments#baseline_aircraft_experiment).

## 7 Conclusions

This overview paper documents the planning, logistics, aircraft capabilities, measurements, manoeuvres and strategies and observations

made under the CLARIFY-2017 deployment of the FAAM aircraft together with complementary NWP and climate modelling studies. Given the wide range of science objectives, and the progress made on these specific objectives, CLARIFY-2017 was an overwhelming success. Key observational findings include:

- The vertical profile of the BBA in the vicinity of Ascension Island has been established to be quite variable with aerosol residing either in the MBL, the residual CBL, or both during the biomass burning season (see also Wu *et al.*, 2020a). Large scale dynamics

and the position of the sub-tropical high appear to have a large control over the levels of BBA in the MBL and residual CBL.

- Biomass burning aerosol size distributions derived from measurements in the residual CBL were found to closely resemble the more limited measurements performed during SAFARI-2000, although a 1-mode or 2-mode model (Peers *et al.*, 2019, 2020; Taylor *et al.*, 2020, Wu *et al.*, 2020a) might be preferred owing to its relative simplicity when utilised in satellite retrieval algorithms.

- State-of-the-art measurement equipment developed since SAFARI-2000 including the SP2 and EXSCALABAR instruments have

given us a much better idea of the microphysical properties of BBA (Taylor *et al.*, 2020; Wu *et al.*, 2020a). The optical properties of many BBA particles can be best represented by a core of black carbon surrounded by a thick shell of organics and, to a lesser extent, inorganics or by semi-empirical mixing rules.



- The thickness of the shell of organics appears to be much thicker (diameter of shell/core of ~2.5) compared to measurements made with identical instrumentation close to the BBA source (diameter of shell/core ~1) for aerosol less than 30 minutes old (Wu *et al.*, 2020b).

- The BBA was rather more absorbing than the earlier measurements from SAFARI-2000 with a mean dried SSA at 550 nm of around 0.80 in the free-troposphere. The SSA of aerosol in the MBL is higher at around 0.85 at 550nm as it includes a proportion of sea-salt aerosol and a higher proportion of sulphate aerosol. We have more confidence in these values owing to the high accuracy of the photoacoustic spectrometer measurements made by the EXSCALABAR instrument, which are not subject to the high levels of correction from filter-based measurements (Davies *et al.*, 2019; Cotterell *et al.*, 2020).

- Mie scattering theory using simple mixing rules such as volume weighting of refractive indices, or the Maxwell-Garnet mixing rule are not able to simultaneously represent both the mass absorption coefficient and the SSA of the BBA (Taylor *et al.*, 2020). This has implications for how to represent aerosol optical properties in global climate models that are fully consistent between the chemical and optical properties.

- The highest resolution LES models utilised here (Herbert *et al.*, 2020) suggest that for semi-direct effects to be of significant magnitude in the region, the separation between BBA and cloud needs to be less than ~500 m. However, studies using larger scale model simulations (Che *et al.*, 2020a) rather contradict this result. Given the differences in horizontal and vertical resolutions between LES and large-scale models, comparing the responses and their BBA-induced and meteorological drivers could potentially solve the apparent disagreement.

- The Met Office operational forecast model (spatial resolution ~11km) was able to capture the variations in the vertical distribution of the BBA reasonably accurately, suggesting that it is a suitable tool for examining aerosol-radiation, aerosol-cloud interactions and fast-feedback processes (Gordon *et al.*, 2018; 2020; Che *et al.*, 2020a). The exception to this was during the POC event that was poorly represented by the model (Abel *et al.*, 2020).

- The coarser resolution UKESM1 climate model that incorporates GLOMAP-mode appears to be able to represent aerosol direct and indirect effects (Che *et al.*, 2020a,b) with reasonable fidelity, which shows the advantages of the Unified Model framework in which the underlying physics is identical between high resolution and lower resolution simulations.

- The BBA examined during CLARIFY-2017 were universally representative of highly aged BBA aerosols of at least 5-7 days since emission with little if any systematic variation in microphysical and optical properties (Taylor *et al.*, 2020).

- Despite the relatively broad wavebands used by the SEVIRI geostationary sensor, aerosol-radiation interactions derived from a newly developed algorithm were shown to compare favourably to those derived from MODIS provided that water vapour profiles were adequately accounted for (Peers *et al.*, 2019, 2020). The geostationary nature of SEVIRI means that the full diurnal cycle of aerosol radiative effects can be examined.

- Aerosol cloud-interactions determined from a statistical analysis of cloud and aerosol in the region are clear. The reduction in cloud effective radius (Twomey, 1977) in polluted conditions is clearly evident, and two different analyses of precipitation suggest that precipitation is inhibited in polluted clouds suggesting that changes in the cloud droplet size distribution reduce the coalescence efficiency (Albrecht, 1989). Models need to be utilised to disentangle the impact of aerosol effects on cloud liquid water and cloud fraction from natural variability (e.g. Dagan and Stier, 2020).

- CLARIFY-2017 was fortunate enough to be able to make some comprehensive measurements of a POC that evolved with time and passed overhead of Ascension Island (Abel *et al.*, 2020). The limited entrainment of overlying BBA into the MBL under such conditions and the relative frequency of such open cells have implications for understanding aerosol-cloud-interactions.





Despite the relative success of the CLARIFY-2017 campaign, which has addressed many of the key objectives, it is envisaged that additional analyses will be performed by the scientific community using the extensive data set of observations. As such, CLARIFY-2017 provides considerable potential legacy work that can be further exploited in the future.

**Acknowledgements:**

We would like to thank Sarah Taylor and Ed Gryspeerdt for putting together Fig1 at the proposal stage. Rear Admiral Tony Radakin and the Joint Forces Command are thanked for their support for the CLARIFY project. Wing Commander Andy Pittock of the RAF was instrumental in making this project a success through the support given to us during the detachment; the can-do attitude of the RAF was very much appreciated throughout the duration of the deployment. Jonny Hobson and the staff of the Obsidian Hotel and thanked for their hospitality, and Jonny deserves particular thanks for providing us with interesting tours of the flora and fauna of the island. The staff of Air

Task, Avalon Engineering and FAAM are thanked for their thoroughly professional service, before, during and after the deployment. The support from Met Office staff at Ascension Island; Amy Raynor (Senior Met Officer), Emma Sillitoe, John Hill and Katie Tobin was also appreciated. This work was mainly supported by the NERC Large Grant NE/L013584/1. JMH, AJ, FM, MIC and FP were also supported by the Research Council of Norway via the projects AC/BC (240372) and NetBC (244141). P.S. would additionally like to acknowledge funding from the European Research Council (ERC) project (RECAP) under the European Union's Horizon 2020 research and innovation

program with grant agreement 724602 and the NERC NE/P013406/1 (A-CURE) project.

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

1205



| Instrument | Facility | Details | Comment |
|---|---|---|---|
| **Aerosol Microphysics** | | | |
| DMT-SPP200 PCASP-100X | FAAM | Resolved aerosol number: 0.10-3 mm. | Wing mounted |
| TSI 3786 Condensation Particle Counter (CPC) | FAAM | Total number, size > 3 nm. | |
| Scanning Mobility Particle Sizer (SMPS) | UoM | Size resolved number: 20-350 nm. | |
| DMT CCN-200 Dual Channel Condensation Nuclei Counter (CCN) | FAAM | Continuous flow CCN at 2 supersaturations | |
| CVI PCASP | Met O | Resolved aerosol number: 0.10-3 ☐m. | Brechtel inlet |
| CVI TSI 3025 CPC | Met O | Total number of residual particles > 3 nm. | Brechtel inlet |
| DMT-SPP 200 PCASP-100X | Met O | Size resolved number: 0.10-3 ☐m. | EXSCALABAR (cabin) |
| Brechtel Counterflow Virtual Impactor (CVI) | Met O/ NCAS | Aerosol residual chemical composition | Brechtel 1204 Inlet |
| DMT UHSAS Ultra-High Sensitivity Aerosol Spectrometer | UoM | 50 – 1000nm aerosol size distribution | Brechtel inlet |
| TSI 3321 Aerosol Particle Sizer (APS). | UoM | 0.5 – 20µm size distribution | Brechtel inlet |
| GRIMM Optical Particle Counter | UoM | 0.25 – 32µm size distribution | Brechtel inlet |
| **Aerosol Composition and Optical Properties** | | | |
| Compact Time of Flight Aerosol Mass Spectrometer (ToF-AMS) | UoM | Size resolved non-refractory 50-700 nm aerodynamic diam. | |
| Single Particle Soot Photometer (SP2) | UoM | Single particle soot detection by laser induced incandescence | |
| Filters | UoM | Sub and super-micron nucleopore | |
| TSI 3563 Nephelometer | FAAM | Scattering coefficient at 450, 550 and 700 nm. | |
| Radiance Research Particle Soot Absorption Photometer (PSAP) | FAAM | Absorption coefficient at 530☐nm. | |
| EXSCALABAR photoacoustic spectrometer (PAS) | Met O | Dry absorption coefficient at 405, 514, 658 nm, thermally denuded at 405 and 658nm | EXSCALABAR PAS, CRDS, PCASP & TAP < 1.3 µm diameter impactor applied to remove coarse aerosol. |
| EXCALABAR Cavity Ring-Down Spectrometer (CRDS) | Met O | Dry extinction coefficient at 405, 658 nm. 75% and 90% RH extinction coefficient at 405 nm. | |
| EXSCALABAR Tri-Absorption Photometer (TAP) | Met O | Dry absorption coefficient at 467, 528 and 652 nm. | |
| **Cloud Physics** | | | |
| DMT Cloud Droplet Probe Cloud-coarse aerosol (CDP-2) & BCPOL | FAAM | Size resolved number: 2☐m to 50 µm, 1 Hz, polarisation | Wing mounted |
| DMT Clouds aerosol and precipitation probe (CAPS CIP-15, CAS) | UoM | Size resolved number: 0.5☐m to 960 µm, 1 Hz | Wing mounted |
| SPEC 2D-S cloud-drizzle spectrometer (2D-S) | UoM | Size resolved number: 10-128000 µm, 100 Hz | Wing mounted |
| SPEC FCDP/FFSSP (ultra-fast cloud droplet spectrometer) | UoM | Size resolved number: 1-60 µm, 50 Hz | Wing mounted |
| Cloud Imaging Probe-15: CIP-15 | FAAM | Size resolved number: 15-960 µm, 1 Hz | Wing mounted |
| Cloud Imaging Probe-100: CIP-100 | FAAM | Size resolved number: 100 mm – 6.4 mm. | Wing mounted |
| Nevzerov hot wire probe | FAAM | Liquid and Total Water Content (LWC & TWC) Liquid and | |
| SEA hot wire probe | FAAM | Total Water Content (LWC & TWC) | |
| **Radiation/Remote Sensing** | | | |
| Leosphere lidar – EZALS450 | Met O | 355nm UV backscattering lidar with depolarisation | |
| Eppley broad band radiometers (BBRs) | Met O | Upper and lower (0.3-3.0 ☐m) and (0.3-0.7 ☐m) fluxes. | Broad-band Pointable |
| Shortwave Spectrometer (SWS) | Met O | Resolved radiances 300-1700 nm at 3-6 nm resolution. | |
| Shortwave Hemispheric Integrating Measurement System (SHIMS) | Met O | Resolved irradiances 300-1700 nm at 3-6 nm resolution. | |
| MARRS | Met O | 89 and 157GHz for Liquid water path retrievals | |
| **Trace Gas Chemistry** | | | |



| Aero-Laser AL5002 | FAAM | CO | ±2.8 ppb @ 1 Hz |
| Teco 49 | FAAM | O3 by UV photometry | Sens 1 ppb@60s |
| Los Gatos Research Inc fast greenhouse gas analyser | FAAM | CO2 and CH4 | ±1.28ppb/0.17 ppm@1Hz |
| TEi43C | FAAM | SO2 | ±0.1 ppb@60s |
| **Thermodynamics** | | | |
| Rosemount Temperature Sensors | FAAM | True Air Temperature 32 Hz | De-iced, Non-deiced housing |
| Chilled Mirror dew point hygrometer? | FAAM | Dew point ~1Hz? | |
| Total Water probe | Met O | Total water content, 64 Hz | |
| WVSS-II: Water vapour | Met O | Water vapour content, 0.4 Hz | |
| Dropsonde systems | FAAM | Profile of temperature, humidity, wind | |
| AIMMS | Met O | 3-D winds, 20 Hz | |
| Turbulence probe | FAAM | 3-D winds, 32 Hz | |

**Table 1: Summary of instruments of major relevance to detachment. FAAM - Facility for Airborne Atmospheric Measurements, Met O - Met Office, UoM - University of Manchester, NCAS - National Centre for Atmospheric Science. Size classifications for cloud/particle distributions are given in diameter.**





| Flight | Date | Take-off | Duration | Sonde # | MODIS overpass | Objectives | BBA (M, C) | AOD 500nm |
|---|---|---|---|---|---|---|---|---|
| C028 | 16/08 | 09:07 | 3:46 | 1 | T (11:10) | Shakedown, aircraft GPS inertial navigation equipment calibration. | M | *0.15* |
| C029 | 17/08 | 08:56 | 3:23 | 2 | T (11:50) | Investigate sharp gradients in AOD forecast by NWP model | M | *0.14* |
| C030 | 17/08 | 14:13 | 3:33 | 1 | A (14:45) | as above | M | |
| C031 | 18/08 | 11:59 | 3:43 | 0 | T (10:55) | Intercomparison with the NASA P3 aircraft (ORACLES) | M | **0.16** |
| C032 | 19/08 | 10:01 | 3:43 | 0 | T (11:40) | Precipitating convective cloud/aerosol interaction, ARI and ACI | M | **-** |
| C033 | 22/08 | 08:54 | 3:45 | 2 | - | Aerosol, radiation & cloud in clean MBL, ARI and ACI | C | **0.27** |
| C034 | 23/08 | 09:02 | 3:29 | 3 | T (11:15) | Mixing of aerosols from residual CBL into MBL, ACI | C | **0.31** |
| C035 | 23/08 | 14:06 | 3:36 | 1 | A (14:05) | Cloud-free direct radiative effect, ARI | C | |
| C036 | 24/08 | 09:03 | 3:02 | 1 | T (11:55) | ACI and ARI of stratocumulus with overlying BBA | C | **0.22** |
| C037 | 24/08 | 13:46 | 3:07 | 2 | A (14:50) | Cloud-free direct effect, ARI | C | |
| C038 | 25/08 | 09:00 | 3:49 | 1 | T (11:00) | ARI/ACI measurements in coordination with MISR | M&C | **0.20** |
| C039 | 25/08 | 14:17 | 3:06 | 0 | - | ARI/ACI in coordination with AMF/LASIC | C | |
| C040 | 26/08 | 08:55 | 3:29 | 0 | T (11:45) | ASI->Monrovia, Liberia, lidar mapping | M&C | *0.40* |
| C041 | 26/08 | 14:14 | 3:05 | 2 | A (14:35) | Monrovia, Liberia-> ASI, lidar mapping | M&C | |
| C042 | 28/08 | 08:55 | 3:28 | 1 | T (11:35) | Above cloud ARI in coordination | M&C | **0.54** |
| C043 | 28/08 | 13:49 | 3:33 | 1 | A (14:25) | Above cloud and cloud free ARI | M&C | |
| C044 | 29/08 | 08:54 | 3:50 | 1 | T (10:30) | Characterisation of MBL cloud and aerosol upwind of AMF | M&C | **0.41** |
| C045 | 29/08 | 14:10 | 3:06 | 1 | A (15:10) | As above | M&C | |
| C046 | 30/08 | 08:45 | 4:06 | 2 | T (11:20) | ACI and ARI | M&C | **0.28** |
| C047 | 01/09 | 08:56 | 2:50 | 1 | T (11:10) | MBL cloud and aerosol characterisation, ARI/ACI | C | **0.37** |
| C048 | 01/09 | 13:26 | 3:57 | 1 | A (14:00) | As above | C | |
| C049 | 02/09 | 08:56 | 3:43 | 1 | T (11:50) | ARI across a cloud boundary | C | **0.32** |
| C050 | 04/09 | 13:28 | 3:46 | 1 | A (14:30) | ACI/ARI coincident with CALIPSO | M&C | **-** |
| C051 | 05/09 | 08:58 | 3:14 | 1 | T (10:45) | Cloud and aerosol measurements upwind on the AMF/LASIC | M&C | **0.34** |
| C052 | 05/09 | 14:09 | 3:29 | 7 | A (15:15) | Across boundary into POC | C | |





| C053 | 06/09 | 08:53 | 3:53 | 3 | T (11:25) | Investigation the transition into POC overhead Ascension Island | C | - |
| C054 | 06/09 | 14:22 | 3:26 | 3 | A (14:20) | POC to SE of Ascension Island | C | |
| C055 | 07/09 | 13:49 | 3:44 | 1 | A (15:00) | Aerosol-Radiation with CALIPSO (but CALIPSO down owing to solar storm). | M&C | **0.20** |

**Table 2: Showing details of the flights performed during CLARIFY-2017 including the take-off and landing times (GMT). Shading is used to indicate flights that were part of a double-flight i.e. both am and pm. M refers to BBA positions in the MBL while C refers to BBA in the residual continental boundary layer. MODIS overpasses and timings are also shown (A=Aqua, T=Terra). The mean daily AOD at 500nm from AERONET stations are shown in italics while those obtained from the hand-held microtops sunphotometers are shown in bold. MBL=marine boundary layer, CBL=continental boundary layer, MISR = Multi-angle Imaging SpectroRadiometer, CALIPSO=Cloud-Aerosol Lidar and Infrared Pathfinder Satellite Observations, POC=pocket of open cells.**



| | Mode 1 (accumulation) | Mode 2 (accumulation) | Mode 3 (coarse) | Ref indices 550nm | SSA 550nm | Comments |
|---|---|---|---|---|---|---|
| **Haywood et al., 2003** | $r_{n1} = 0.12\ \mu m$ <br> $\sigma_1 = 1.3$ <br> $N_1 = 0.996$ | $r_{n2} = 0.26\ \mu m$ <br> $\sigma_2 = 1.5$ <br> $N_2 = 0.0033$ | $r_{n3} = 0.617\ \mu m$ <br> $\sigma_3 = 2.23$ <br> $N_3 = 0.0007$ | 1.54 - 0.018i | 0.88 (0.91*) | Volume weighting Off Namibian coast. Optimised to represent 550nm optical parameters |
| **Haywood et al., 2003** | $r_{n1} = 0.117\ \mu m$ <br> $\sigma_1 = 1.25$ <br> $N_1 = 0.9997$ | $r_{n2} = 0.255\ \mu m$ <br> $\sigma_2 = 1.5$ <br> $N_2 = 0.0033$ | **n/a** | 1.54 - 0.018i | 0.87 (0.90*) | As above Vicinity of Ascension Island |
| **Peers et al. 2019, 2020;** | $r_{n1} = 0.119\ \mu m$ <br> $\sigma_1 = 1.42$ <br> $N_1 = 0.999631$ | **n/a** | $r_{n2} = 0.617\ \mu m$ <br> $\sigma_2 = 2.23$ <br> $N_2 = 0.000369$ | 1.51 — 0.027i | 0.85 | Volume weighting Optimised for SEVIRI wavelengths. Residual CBL. |
| **Wu et al., 2020** | $r_{n1} = 0.116\ \mu m$ (CBL) <br> $\sigma_1 = 1.46$ (CBL) <br> $r_{n1} = 0.101\ \mu m$ (MBL) <br> $\sigma_1 = 1.45$ (MBL) | **n/a** | **n/a** | 1.54 — 0.029i (CBL) | 0.83CBL, 0.85MBL | Following work by Peers et al. (2019). |

**Table 3: Showing the models that have been fitted to aerosol size distributions. *Originally a SSA at 550nm of 0.91 (Ascension Island) and 0.90 (Off coast of Namibia) was reported for BBA, but this was reassessed using more rigorous corrections to absorption and scattering measurements to yield 0.88/0.87 by Johnson et al., 2008. $N_x$ represents the fractional number concentration in mode x. The refractive indices represent the effective refractive indices that combine with the size distribution and Mie scattering theory to yield the reported SSA.**





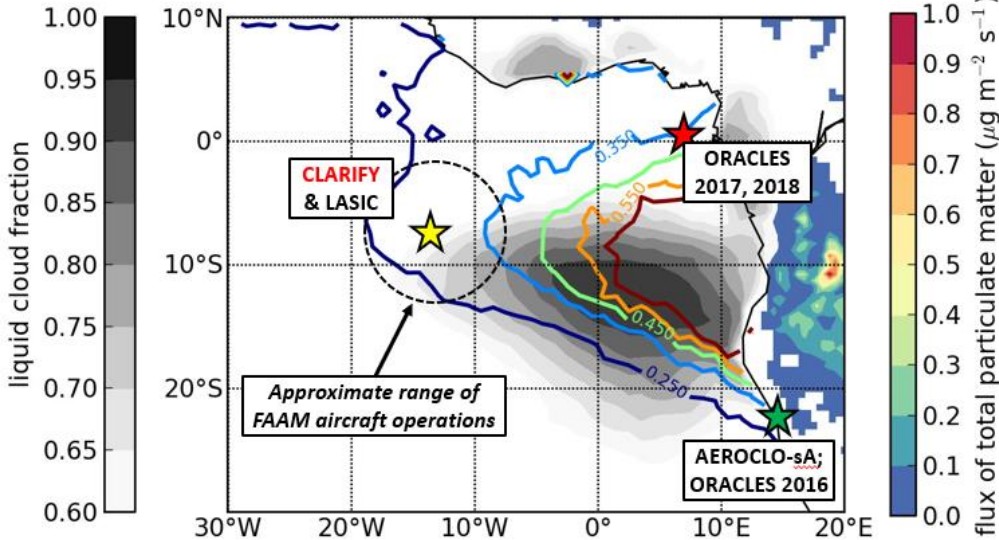

**Figure 1: 2003-2011 mean Aug-Oct AODs (coloured contours) retrieved from the MODIS satellite, MODIS cloud fraction (black and white colour scale), and Global Fire Emissions Dataset (GFED) aerosol emission estimates (colours over land). The yellow star shows the position of Ascension Island with a dashed circle representing the approximate operating range of the FAAM aircraft. The position of São Tomé where ORACLES operations were performed, and Walvis Bay where AEROCLO-sA operations were performed are marked by red and green stars respectively.**





**Figure 2: The annual mean direct radiative forcing (aerosol-radiation-interaction) of BBA calculated by 16 different AEROCOM models. Units W m⁻². The most negative radiative forcing is in the top left hand corner while the most positive radiative forcing is the bottom right hand corner. Reproduced after Zuidema *et al.*, 2016.**



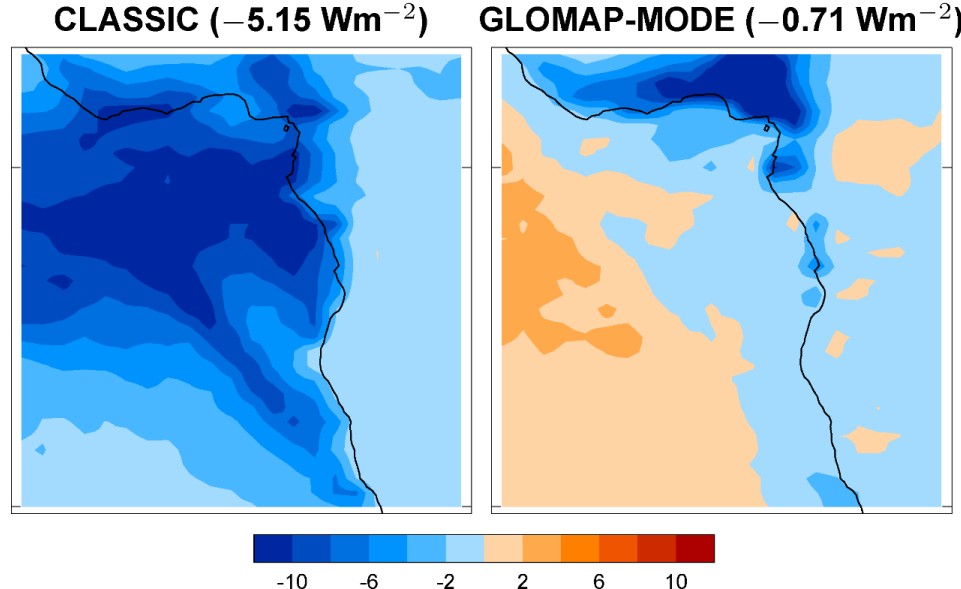

**Figure 3: The annual mean aerosol indirect effect (cloud-aerosol-interaction) diagnosed for two aerosol schemes (CLASSIC and GLOMAP-MODE) within the HadGEM2 climate model. Units (W m⁻²).**

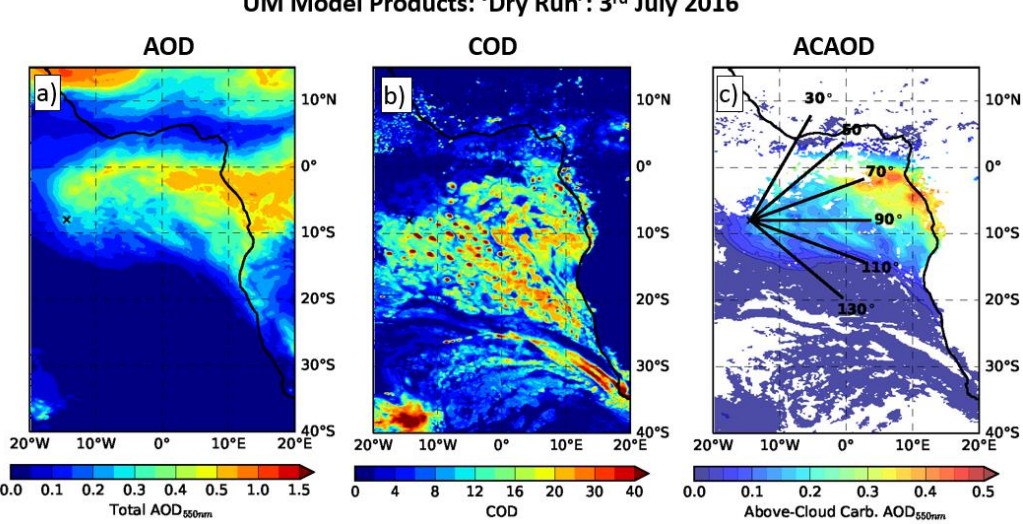

**Figure 4: Model forecasts with the CLUMP model for a wavelength of 550nm for a) aerosol optical depth (AOD), b) cloud optical depth (COD), c) above cloud AOD (ACAOD). The solid lines on (c) indicate transects over which vertical-horizontal distribution maps aerosol and cloud properties were provided as data products, some of which are provided in Figure 5.**

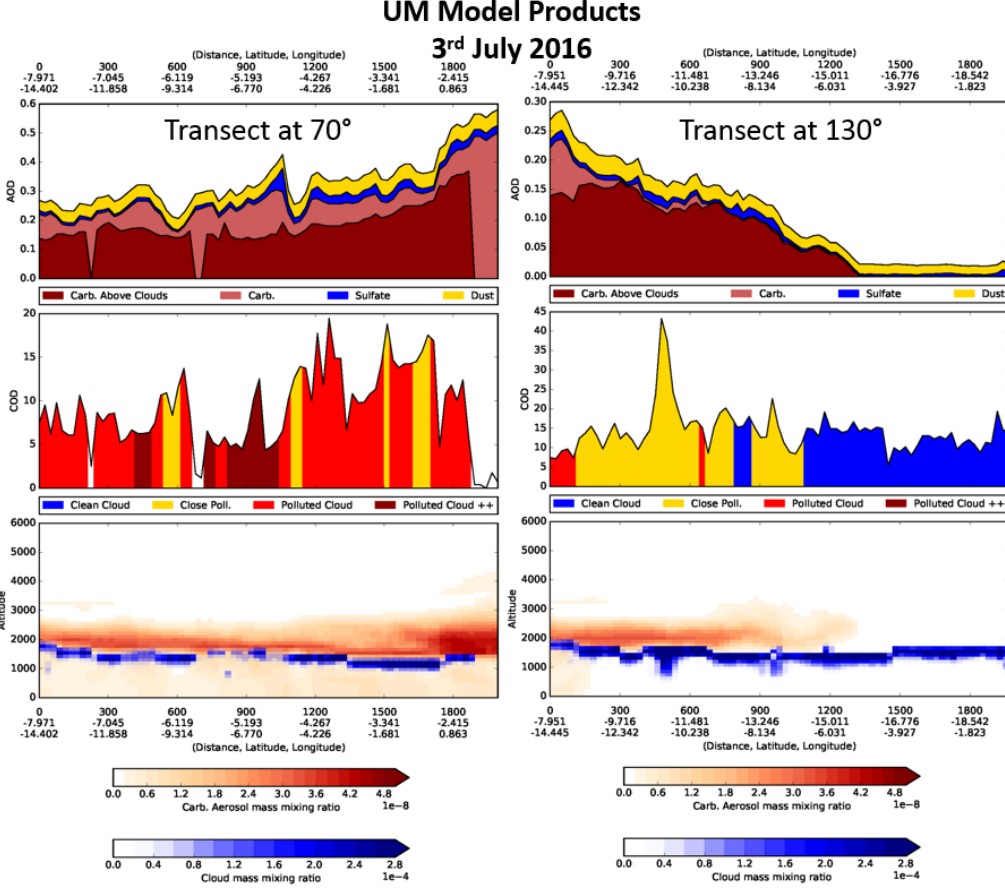

**Figure 5: Showing model products derived at the 70°transect (column 1) and 130°transect (column 2) shown in Figure 4. Aerosol optical depth (AOD) is split into the three component aerosol types within the CLUMP model and the carbonaceous aerosols that include BBA are further sub-divided into above cloud and below cloud components. Cloud optical depth (COD) is sub-divided into i) 'clean' (aerosol concentrations < 3 µg kg⁻¹), ii) clean, but close to pollution (aerosol concentrations < 3 µg kg⁻¹ but within 200m in the vertical of aerosol ≥ 3 µg kg⁻¹), iii) polluted (aerosol concentrations ≥ 3 µg kg⁻¹), iv) very polluted (aerosol concentrations ≥ 10 µg kg⁻¹). The third row shows the vertical profile of carbonaceous aerosols (red) and cloud (blue) mass mixing ratios.**





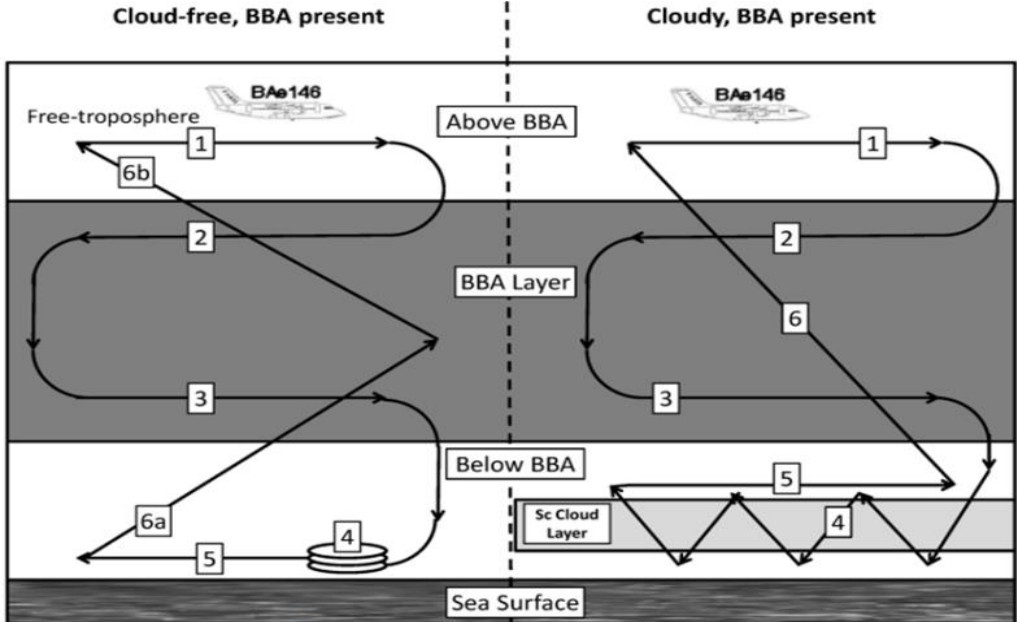

**Figure 6: Schematic diagram of the manoeuvres that were typically performed during cloud-free and cloudy conditions. The numbers marked on the schematic represent the manoeuvres referred to in the text.**

1260

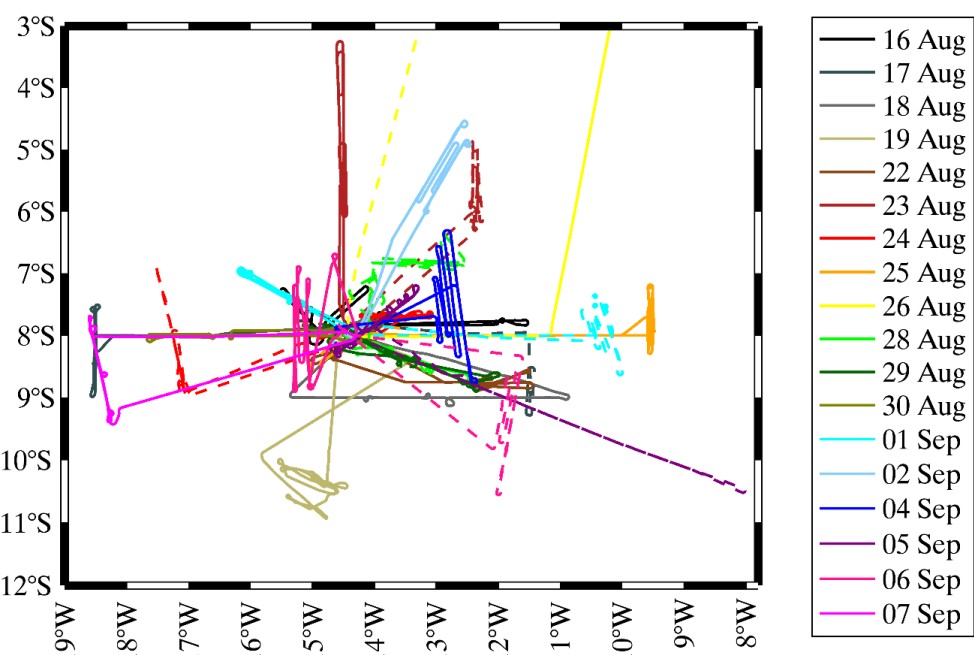

**Figure 7: The geographical position of the sorties that were performed during CLARIFY-2017.**





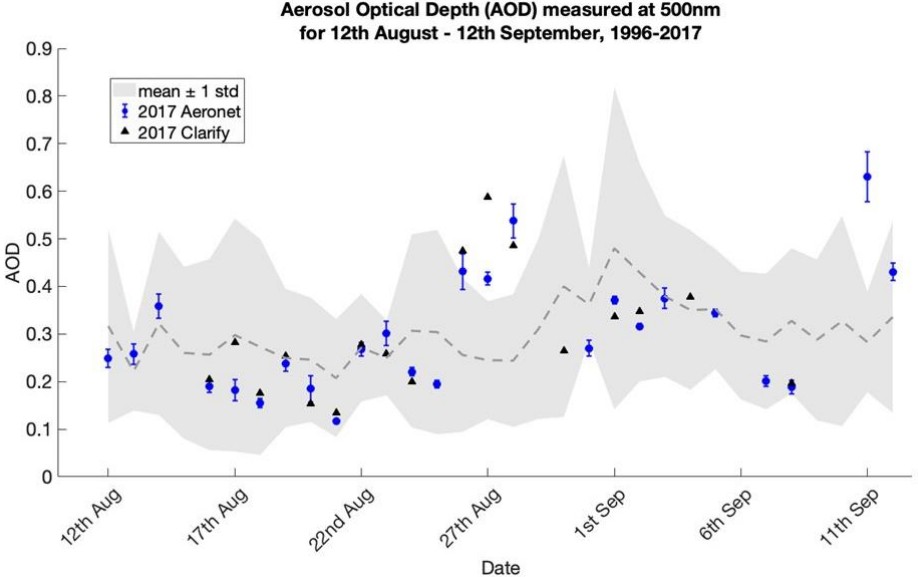

**Figure 8: AERONET and CLARIFY daily average AOD data, measured at 500nm, plotted for the period 12th August - 12th September, 2017. The data from each year is represented by a different colour, with 2017 data shown in black circles. The CLARIFY data, measured with a Microtops sun photometer, is represented by the black triangles. The mean of data between 1996-2016 is shown by the dashed line, with the grey shaded area representing ± one standard deviation. The vertical lines on the ARM AOD data shows ± one standard deviation of this data.**

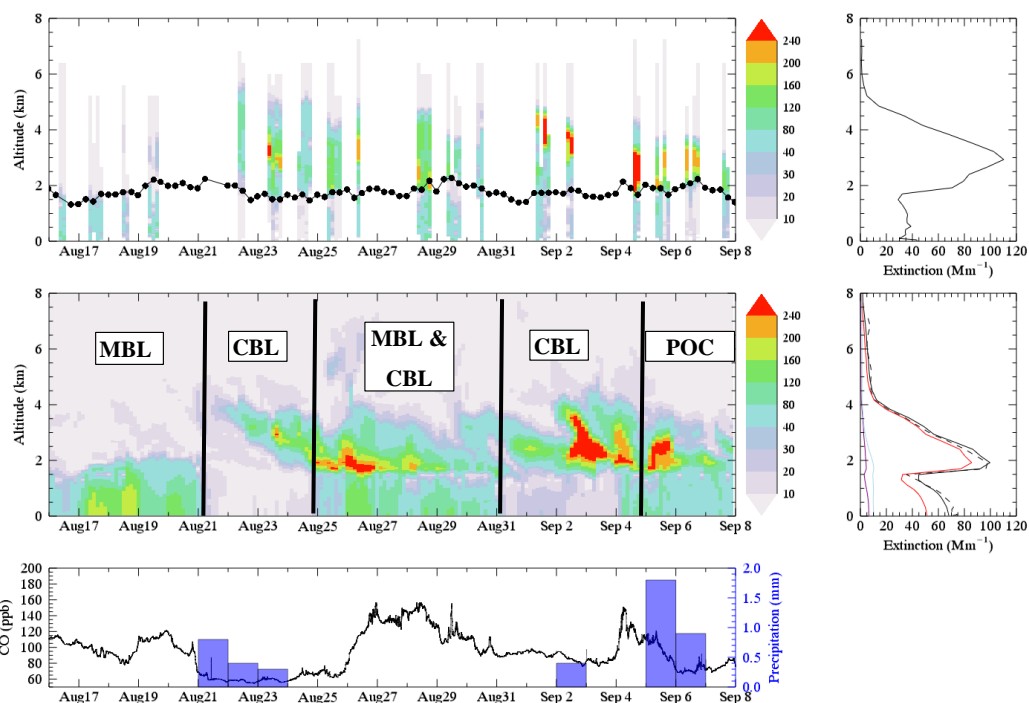

**Figure 9: Time series plots showing from top to bottom i) vertical profiles of the sub-micron aerosol extinction (x10⁻⁶ m⁻¹) derived from the EXSCALABAR instrument (405 nm, dry), with radiosonde estimates of the MBL inversion height overlaid with black circles. A mean vertical profile of aerosol extinction is also shown in the right-hand panel. ii) vertical profiles of aerosol extinction (x10⁻⁶ m⁻¹, 405nm, dry) from the Met Office CLUMP forecast model; MBL and CBL and POC are used to discriminate the vertical profile regimes described in Table 2. The right-hand panel shows the mean profile of BBA aerosol only in red, industrial aerosol is shown in cyan and mineral dust in blue. The black solid and**





dashed lines show the mean aerosol profile (dry) and the mean profile (subsampled when there were FAAM flights) for all CLUMP aerosol components iii) the carbon monoxide (CO) concentrations measured at the AMF on Ascension Island at ~330 m ASL in the MBL, with precipitation measured at the Met Office on Ascension Island included.

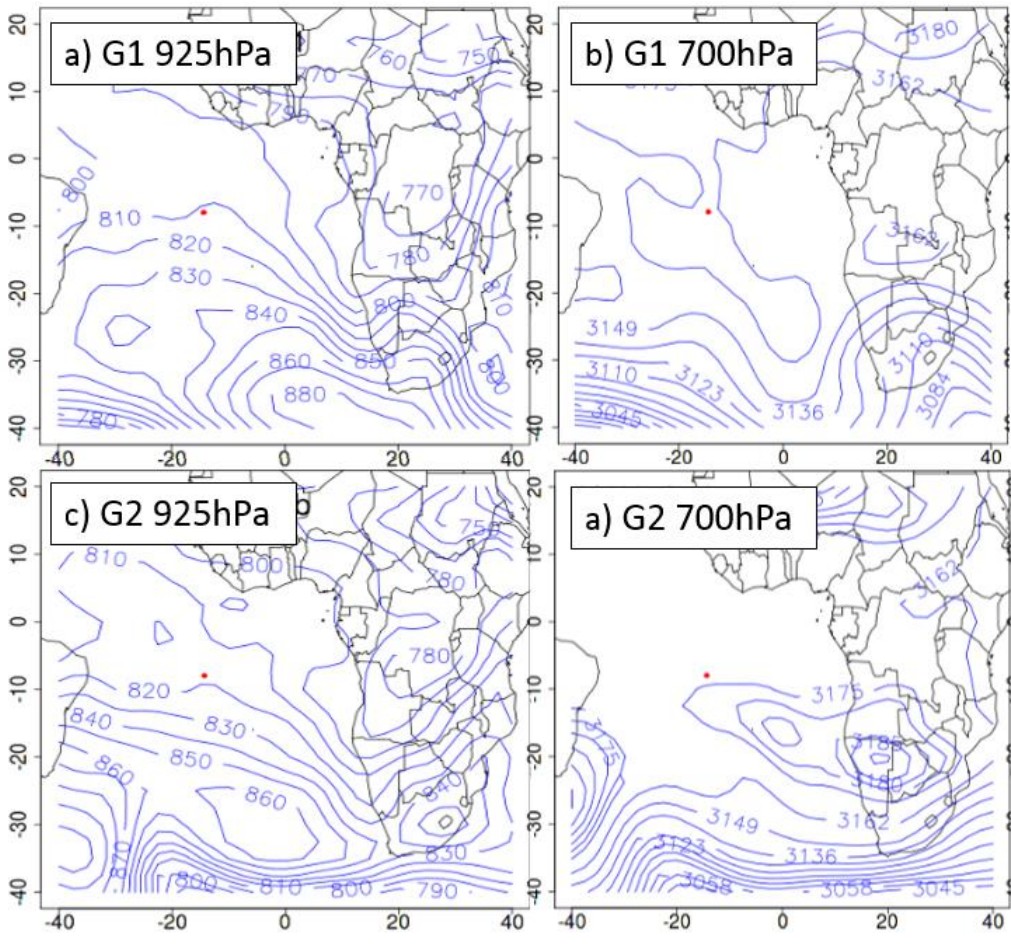

1280

**Figure 10: Mean geopotential height (m) of (a) Group G1 at 925 hPa, (b) Group 1 at 700 hPa, (c) Group G2 at 925 hPa, (d) Group G2 at 700 hPa. The x-axis and y-axis are longitude and latitude in degree, respectively. Ascension Island is marked as a red dot.**


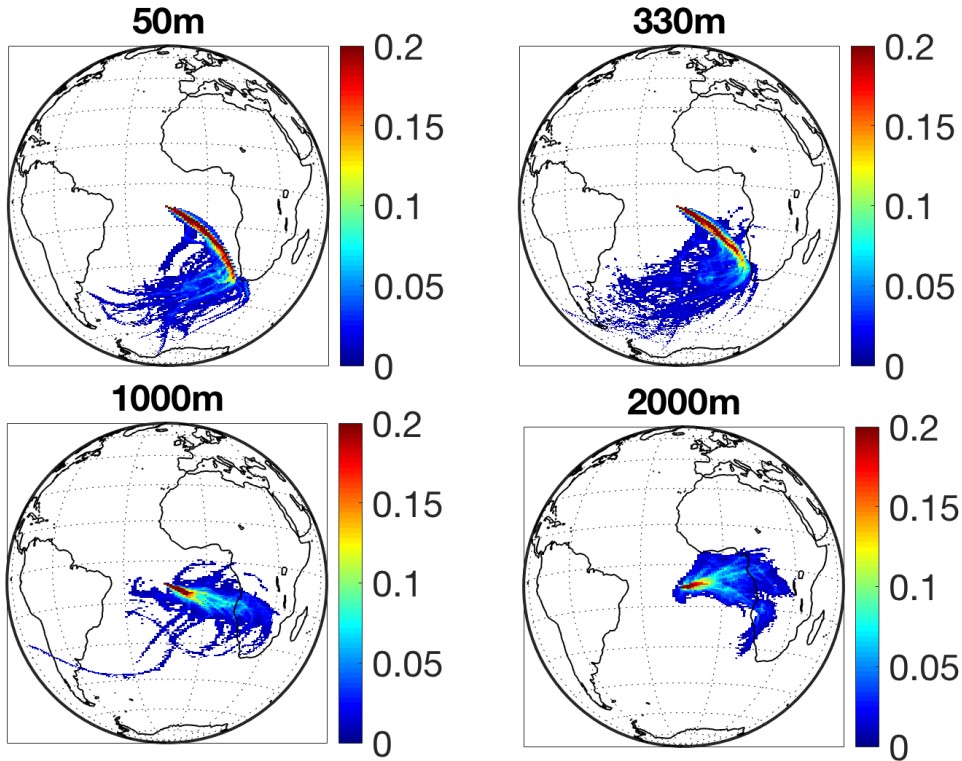

**Figure 11: Trajectory density plots for starting heights 50m, 330m, 1000m, and 2000m above terrain height using HYSPLIT (Stein *et al.*, 2015) 10 day back trajectories, averaged over August 2017. There is one trajectory initiated each hour. The colour-bar depicts the density of trajectories over each 0.5°x1° latitude-longitude grid cell, with each grid cell having minimum of 5 trajectories passing through it and is displayed as a relative area weighted frequency.**

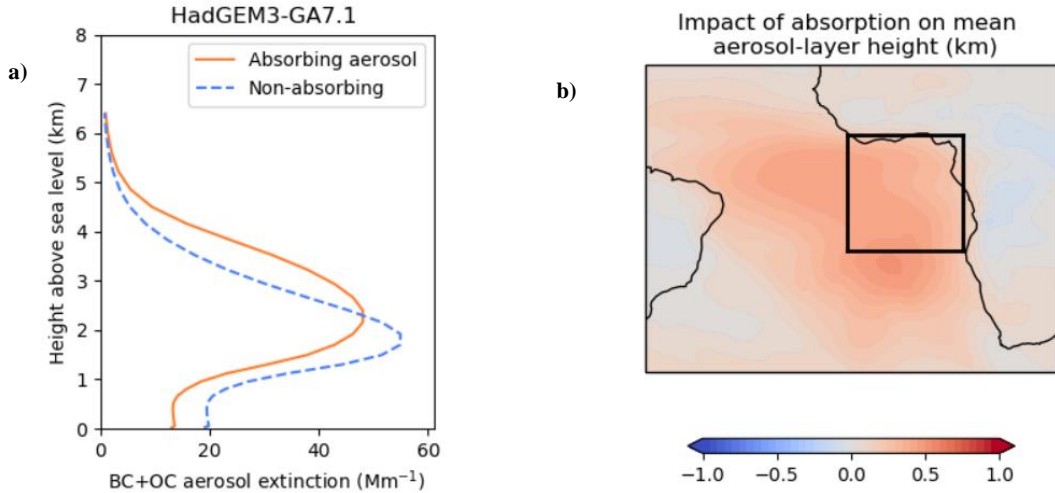

**Figure 12: Showing a) the increase in the mean altitude of BBA in the vertical profile in HadGEM3-GA7.1 version of the climate model (atmosphere only) for August/September over the area shown by the box in Figure b; b) the change in the mean altitude of the aerosol loading.**





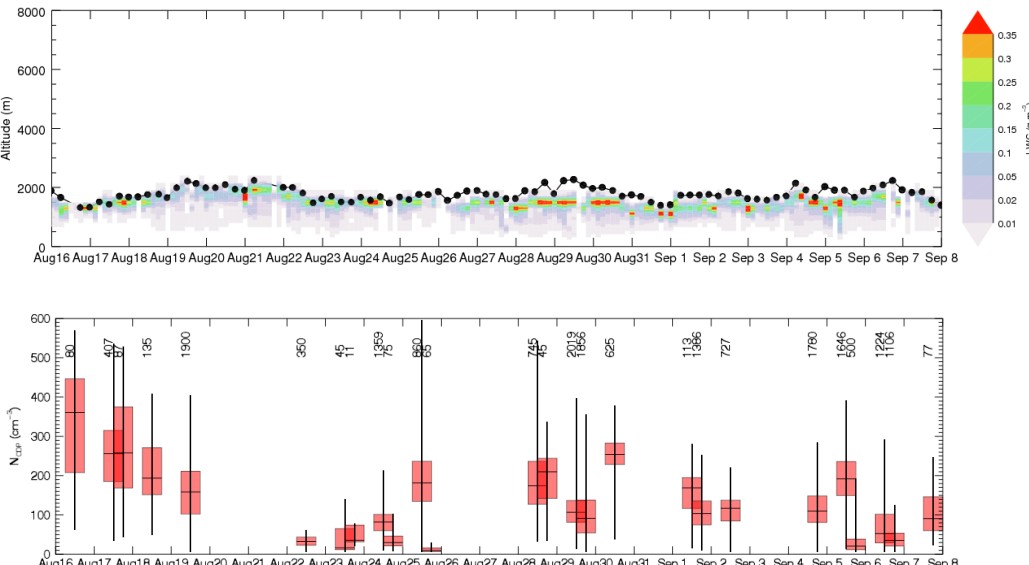

**Figure 13: (i) Vertical profiles of cloud LWC from the Met Office CLUMP model (colour scale) and the position of the MBL inversion derived from radiosonde ascents (black circles). (ii) box and whisker plots showing the cloud drop number concentration measured from the CDP on each flight. The median value is shown by the horizontal black line, the 25 and 75 percentiles by the limits of the boxes and the range is shown by the whiskers. CDP data are selected for points where LWC > 0.05 g m⁻³ and N > 5 cm⁻³. The number of 1 Hz data points that meet these thresholds are displayed on the figure for each flight.**

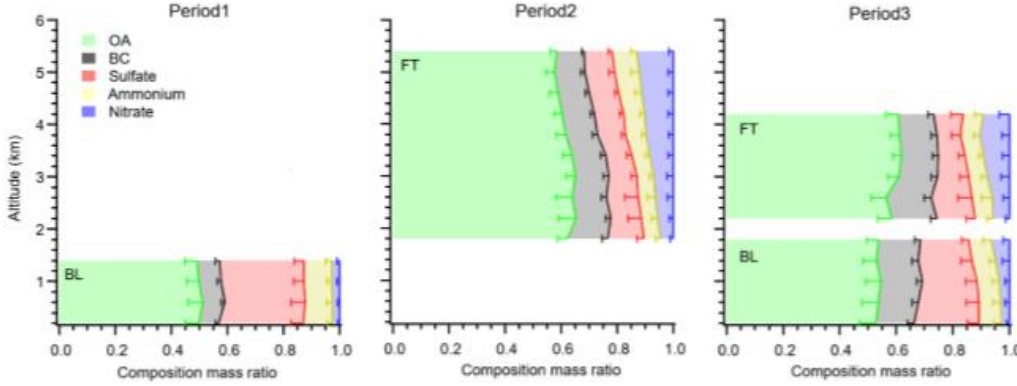

**Figure 14: The average vertical distribution of PM1 chemical composition ratios in the BB-polluted residual CBL and MBL separately in each period. The width of colour bars represents average mass ratio of different species in each 400 m bin. The error bars represent one standard deviation. Period 1 corresponds to BBA in the MBL (16-21 July), period 2 to BBA in the residual CBL (26-31 July) and period 3 to BBA in both (22-25 July, 1-5 July) as per Figure 9.**


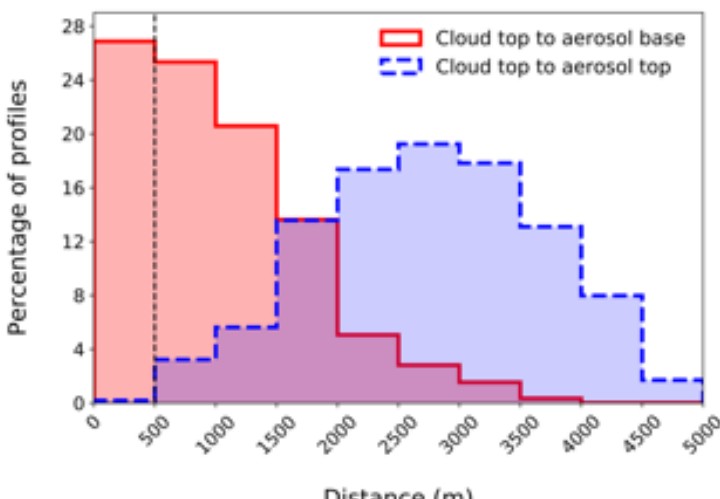

Figure 15: Analysis of the vertical gap that is apparent between underlying clouds and overlying BBA observed using the CATS lidar. The data analysed is for July, August, September in the years 2014-2017. All profiles are within the area 20°S-5°N and 10°W-15°E and are taken from retrievals where there is a single liquid cloud layer below 2.5 km and a single BBA layer in the same profile. The red solid line corresponds to distance between cloud top and BBA base (percentage of occurrence is for all profiles), and the blue dashed line corresponds to the distance between cloud top and BBA top (percentage of occurrence only for profiles where the BBA layer is above the cloud). The black dashed line at 500 m highlights the distance beyond which LES simulations by Herbert et al. (2020) suggest there is little or no SDE.

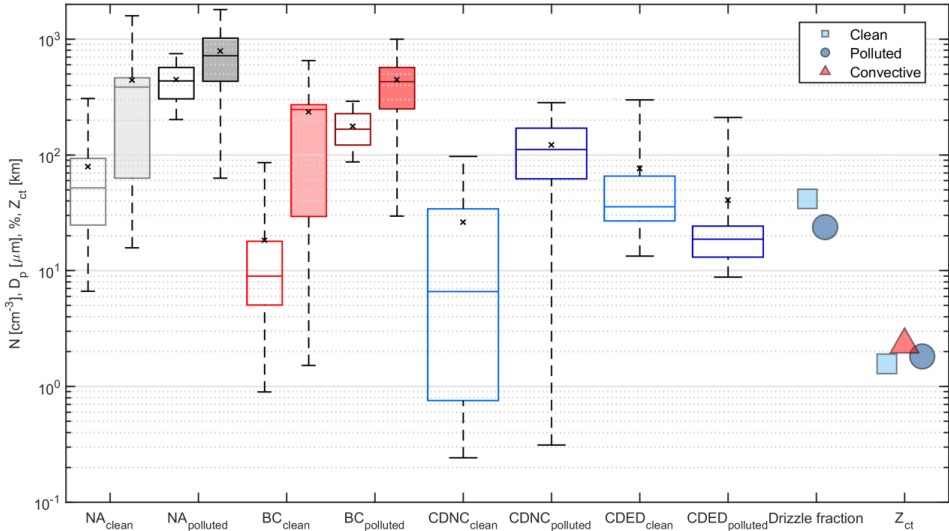

Figure 16: Statistical overview of aerosol and cloud properties. Total aerosol (NA, PCASP) and black carbon (BC, SP-2) concentration data are taken from cloud free conditions (LWC < 0.01 g m⁻³). Aerosol data is shown for conditions broadly representative of the marine boundary layer (Z < 1500 m, unfilled boxes) and free troposphere (2500 > Z > 4000 m, filled boxes). Cloud droplet number concentration (CDNC, CDP) and cloud droplet effective diameter (CDED, composite of CDP and 2DS size distributions) are calculated using a minimum LWC threshold of 0.01 g m⁻³. Drizzle fraction is the ratio of the total number of data points containing drizzle (D > 100 µm, drizzle water content > 0.01 g m⁻³) to in cloud





data points (total water content > 0.01 g m$^{-3}$), expressed as a percentage. Cloud top altitude ($Z_{ct}$) is the average value of cloud top determined
from aircraft profiles. Convective cases ($Z_{ct}$ > 2000 m) are removed from the clean and polluted $Z_{ct}$ averages and displayed separately. All data
has been split into clean (CO < 83 ppb) and polluted (CO > 83 ppb) conditions. Black markers indicate mean; box indicates interquartile range
and median values, whiskers present 5$^{th}$ and 95$^{th}$ percentiles.

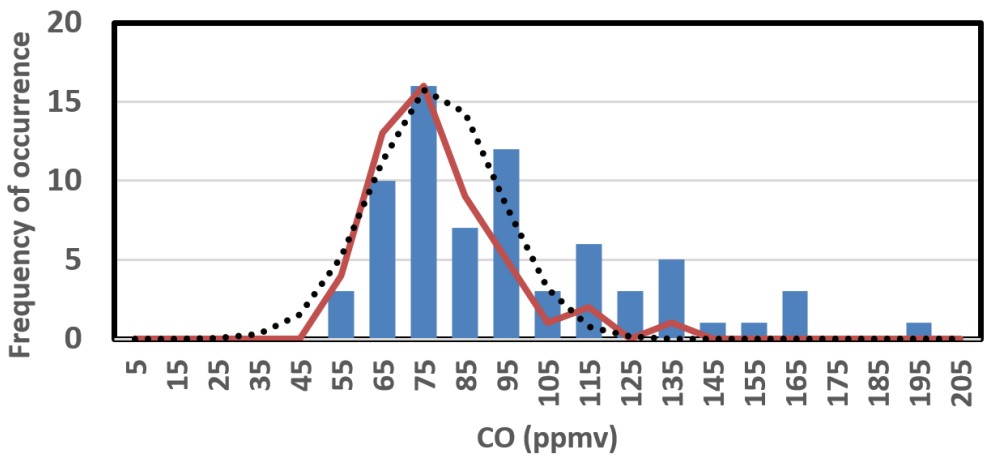

Figure 17: The carbon monoxide (CO) concentrations for precipitating (red curve) and non-precipitating days (blue bars). A Gaussian fit to the
non-precipitating data is shown by the dotted lines.

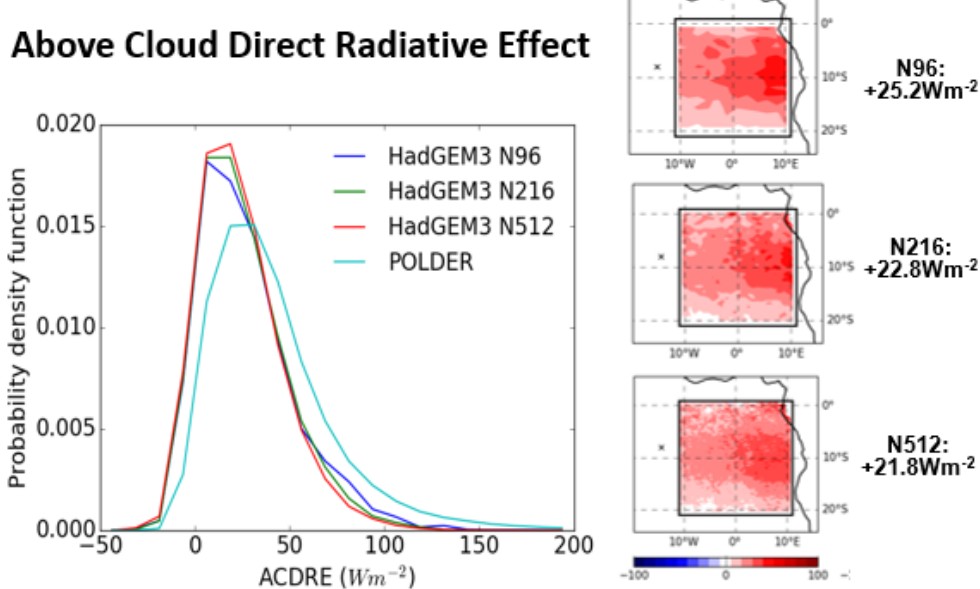

**Figure 18: Showing the above cloud direct radiative effect diagnosed from the Unified model (N96, N216 and N512 resolution, approximately 140, 60 and 25 km respectively) over the area shown in the panels in the right-hand column. The probability density function of the above cloud direct radiative effect is also shown from POLDER after (Peers *et al*., 2016). The intercomparison is for August-September 2006 and model data is matched to instantaneous POLDER retrievals.**
