# Peer review of "Overview: The CLoud-Aerosol-Radiation Interaction and Forcing: Year-2017 (CLARIFY-2017) measurement campaign"

_Atmospheric Chemistry and Physics, 2020_

## Referee Comment (RC1) · Johannes Quaas (Referee) · 27 Oct 2020

Haywood et al. present a comprehensive overview of the CLARIFY-2017 measurement campaign. From the paper, it is evident that this was a diligently planned and well-conducted campaign, that had scientifically important, well-posed objectives and that excellently linked to other international initiatives and to previous campaigns. The results presented in the paper are impressive, it is evident that already at this short time after the campaign it proved highly useful to improve the understanding of clouds, aerosols, and their interactions in the region and beyond it. The manuscript is excellently written and of large interest to the readership of Atmos. Chem. Phys.

[Figure]

I only have a number of specific comments that I suggest the authors address in a revision.

l29 reconstruct past climate (?)

l30 In fact, some in the community would nowadays use the term "natural laboratory" for a situation where a quasi-external aerosol perturbation is imposed (e.g. volcanic eruption), which is not the case here. So maybe the authors want to choose another term.

l38-43 A reformulation would be helpful to make the distinction between aim 1 ("improve . . . model estimates") and 4 ("improve numerical models") more clear.

l42/43 "deployment of the . . . campaign" seems a wrong formulation

l46/53 is 2013 "recent"?

l84-92 perhaps as a bullet list?

l87 close bracket

l118 why the droplet sizes?

l133-138 It would be useful to spell out what the authors have in mind with the "detailed mechanisms of the semi-direct effect". After all, it seems much more straightforward to parameterise than any indirect effect.

l141 as a follow-up question – isn't simply the absorption parameterised and the rest of semi-direct mechanims follows?

l167 one "aerosol scheme" too many in the sentence.

l173 A few words on how the widespread positive forcing due to aerosol-cloud interactions in the GLOMAP version is explained would be useful. It seems a peculiar result, in particular in a stratocumulus region.

l181 what is "Norde-Est"?

l198 maybe the modelling section deserves a new paragraph?

l215/216 It is of course clear these key objectives were carefully formulated and iterated many times, probably since well before the campaign. Nevertheless, I ask myself - How does one "understand" a property? Is "understanding" not usually referring to some causality?

l249 As the data assimilation for the dust sounds very innovative to me, a few more words on what exactly was done would be useful.

l275 is the unit wrong in the PDF (g kg-1)?

l298 Is satellite validation the best scientific goal, or not rather the inverse – using the satellite to put the aircraft measurements into a larger-scale context?

l300 CIMEL without "s"

l335-336 wow

l390 should that be $\mu$m?

l394 $P$ and $N_{ccn}$ taken at both flight level? Or CCN below cloud? (maybe it is wiser to put this analysis description to where the results are shown, rather than providing it as an isolated example in this section).

l405 The acronym POC needs to be explained here.

l427 "solely" is in my opinion an oversimplification.

l455 the ERA5 reference should be updated to Hersbach et al. QJRMS 2020.

l468 A better reference here is Costantino and Bréon 2013.

l505-509 Why not show this key result on aerosol-cloud interactions more clearly, e.g. by a scatter plot of MBL BBA mass or number vs. cloud droplet concentration?

l517 why "also"? at first glance the two statements (vertical gradient in SSA, vertically constant mass absorption coefficient) are in contradiction.

l598 isn't this contradicting the result quoted earlier that SSA increased with height?

l564-617 In my opinion, this is a bit of a lengthy discussion of some details that maybe is not a headline result of CLARIFY (i.e. this section might be shortened).

l692 the *refs* are missing ...

l699-701 the $\pm$ ranges are awkward, as they imply strong possibility of negative values. An asymmetric range is required in my opinion.

l703 $\mu$; are the median values for diameters as in Fig. 16? this should be clarified as it is uncommon (typically radii are reported).

l706 why is this result not clear? Isn't this a large factor? It is also consistent with the results reported for LWP and for Zct.

l774-786 This seems to me a pure modelling study that is disconnected to the observations of CLARIFY. If so, I suggest to drop the discussion. If my impression is wrong, it would be useful to make the link to the observations at this point.

l816-819 I don't remember a detailed discussion of this result in the main body. Is it a key CLARIFY observational result?

l820-824 It would be useful to make the connection to the observations, i.e. that in only a quarter of the cases, a SDE is expected.

l829-831 This seems to be a bit out of the blue, in particular the statement on the indirect effect, especially given the peculiar indirect effect in GLOMAP. A more clear link to the body text and the observations is necessary.

l834-837 Sorry if again I missed a point, but were aerosol-radiation interactions from MODIS shown in the body, and compared to the SEVIRI results?

l853 "are thanked"

l1206/Table 1 is "resolved number" and "size resolved number" the same or something different (if the former, give it the same name, if the latter, explain). some $\mu$ don't appear in the PDF.

l1245 it would be useful to show zero COD as white or grey in panel (b) (presumably white in (c) means no cloud?). Why does the figure legend write "Carb."?

l1265 the "each year different colour" is probably a remnant from a former version of the Figure? "ARM AOD" or rather "Aeronet AOD"?

l1290 The caption for (b) should briefly explain how the mean altitude is calculated/defined.

l1295 why not focus on the lower 3000 m in panel (a) so that the differences between radiosondes and model are better visible?

l1305 The caption needs to clarify where the data is from.
l1330 why not also a Gaussian fit to the precipitating days?

---

## Referee Comment (RC2) · Anonymous Referee #2 · 19 Nov 2020

**Start of review**

The paper provides a very comprehensive overview of the CLARIFY-2017 experiment that took place over the Southeastern Atlantic (SEA) Ocean during the biomass burning season over southern Africa. The paper covers all aspect related to such an intense detachment effort in a remote area, including pre-campaign preparation (incl. dry runs, satellite climatology and model strategy) and interactions with other detachments which took place at roughly the same period in the SEA basin, namely the LASIC, ORACLES and AEROCLO-sA campaigns. The paper also presents some scientific highligths to illustrate the body of new knowledge gained from the CLARIFY project.

Overall, I find the paper to be very clear and well written, and only have minor comments or suggestions to make to the authors.

L49: "Aerosol-cloud interactions, also termed indirect effects, arise from aerosols acting as cloud condensation nuclei (CCN) in warm clouds": Aren't ice clouds concerned too? Should not this be mentionned as well here on the basis of the recent ICE-D detatchment made from Cape Verde?

L156-157: "However, it is difficult to fully discern the level of interaction between clouds and aerosols because of the sensitivity of lidars in the free troposphere (Watson-Parris et al., 2018) and the attenuating effects of a thick layer of aerosols overlying clouds." This is agreed, but the work by Daeconu et al. (this special issue) has shown that this can be managed as long as one uses CALIOP observations at 1064 nm. I think it would be worth mentionning this study:

Satellite inference of water vapour and above-cloud aerosol combined effect on radiative budget and cloud-top processes in the southeastern Atlantic Ocean Lucia T. Deaconu, Nicolas Ferlay, Fabien Waquet, Fanny Peers, François Thieuleux, and Philippe Goloub Atmos. Chem. Phys., 19, 11613–11634, https://doi.org/10.5194/acp-19-1161 3-2019, 2019

368: SLR already defined.

Figure 7: could be improved by showing the type of flying as a function of the objectives...? And show the kind of vertical sampling made during the flights as well.

Figure 10: It would be nice to show winds associated with the composited geopotential patterns.

Figure 11: nice but why have you not composited the back trajectories on the same days as for the G1 and G2 ? why do this on the whole of August?

End of review

**ACPD**
**Discussion** paper

---

## Author Comment (AC1) · 23 Nov 2020

**Response to Reviewers' Comments**

**We would like to thank both reviewers for their efforts in reviewing the manuscript. It is quite an undertaking to review these lengthy overview papers, so their efforts are very much appreciated.**

**Reviewer #1: Johannes Quaas.**

We would like to thanks Johannes Quaas for his thorough review of the manuscript. We are pleased that Johannes has found that CLARIFY-2017 was considered a "diligently planned and well-conducted campaign, that had scientifically important, well-posed objectives and that excellently linked to other international initiatives and to previous campaigns". We have accounted for Johannes' comments as follows:-

l29 reconstruct past climate (?)

Agreed – amended.

l30 In fact, some in the community would nowadays use the term "natural laboratory" for a situation where a quasi-external aerosol perturbation is imposed (e.g. volcanic eruption), which is not the case here. So maybe the authors want to choose another term.

Agreed – amended to optimal region.

l38-43 A reformulation would be helpful to make the distinction between aim 1 ("improve . . . model estimates") and 4 ("improve numerical models") more clear.

The distinction between I) and iv) has been improved by simply removing "and their" in item iv), meaning that i) refers to improving representation of ARI and ACI and iv) improving the impacts on weather and climate in numerical models.

l42/43 "deployment of the . . . campaign" seems a wrong formulation

We think this is OK as is

l46/53 is 2013 "recent"?

Agreed that this is subjective. Therefore changes recent -> most recent

l84-92 perhaps as a bullet list?

Agreed that it looks better as a bullet list.

l87 close bracket

ammended

l118 why the droplet sizes?

We think this is OK – the cloud optical depth is proportional to the LWP and inversely proportional to the droplet effective radius. We could have stated the COD instead, but prefer to convolute the LWP and the effective radius in this sentence.

l133-138 It would be useful to spell out what the authors have in mind with the "detailed mechanisms of the semi-direct effect". After all, it seems much more straightforward to parameterise than any indirect effect.

Agreed. Sentence removed.

l141 as a follow-up question – isn't simply the absorption parameterised and the rest of semi-direct mechanims follows?

See response to the above – we agree.

l167 one "aerosol scheme" too many in the sentence.

Agreed – removed.

l173 A few words on how the widespread positive forcing due to aerosol-cloud interactions in the GLOMAP version is explained would be useful. It seems a peculiar result, in particular in a stratocumulus region.

Agreed. This was actually a combination of the aerosol-indirect effect and the aerosol-semi-direct effect. The positive forcing is caused by a very small reduction in the cloud fraction in the model with GLOMAP mode.

We have added this sentence and amended the caption accordingly:- "Therefore, when combined with a slight reduction in the cloud fraction associated with the aerosol-semi direct effect, some areas of the south Atlantic shown in Figure 3b are subject to a positive radiative effect."

l181 what is "Norde-Est"?

Thanks for bringing this to our attention – we had spelled it incorrectly, but it is an official region of Brazil. Granted – it does mean North East, but it specifically contains 9 smaller states. See https://en.wikipedia.org/wiki/Northeast_Region,_Brazil#:~:text=Nordeste%20stretches%20from%20the%20Atlantic%20seaboard%20in%20the,exploration%2C%20settlement%20and%20economic%20development%20of%20the%20region.

l198 maybe the modelling section deserves a new paragraph?

OK

l215/216 It is of course clear these key objectives were carefully formulated and iterated many times, probably since well before the campaign. Nevertheless, I ask myself - How does one "understand" a property? Is "understanding" not usually referring to some causality?

OK – some of the text has been imported from the original proposal which had a very strong word constraint. A few more words have been added to clarify the objectives.

l249 As the data assimilation for the dust sounds very innovative to me, a few more words on what exactly was done would be useful.

Mineral dust assimilations has been performed by a number of studies. We now include reference to Pope et al. (2016) and O'Sullivan et al. (2020) so that the reader can follow this up more easily should they require.

l275 is the unit wrong in the PDF (g kg-1)?

These are the correct thresholds.

l298 Is satellite validation the best scientific goal, or not rather the inverse – using the satellite to put the aircraft measurements into a larger-scale context?

Good point. We have added this to the end of the sentence.

l300 CIMEL without "s"

Corrected.

l335-336 wow

We agree! One of the things that we have to do is prove outreach and this proves it very nicely we think.

l390 should that be μm?

No – there are multiple probes capturing the size distribution of aerosols, cloud droplets and precipitation particles up to mm sizes. Further details can be found in Table 1.

l394 P and Nccn taken at both flight level? Or CCN below cloud? (maybe it is wiser to put this analysis description to where the results are shown, rather than providing it as an isolated example in this section).

We agree. This detail does somewhat stand out as an isolated example and probably gives too much detail compared to other aspects. We have removed it.

l405 The acronym POC needs to be explained here.

Agreed.

l427 "solely" is in my opinion an oversimplification.

Agreed. Have changed to predominantly.

l455 the ERA5 reference should be updated to Hersbach et al. QJRMS 2020.

Thank you. Amended as suggested.

l468 A better reference here is Costantino and Bréon 2013.

OK – changed.

l505-509 Why not show this key result on aerosol-cloud interactions more clearly, e.g. by a scatter plot of MBL BBA mass or number vs. cloud droplet concentration?

This is a little bit of a tricky one. The main results are to be shown in the associated detailed papers (e.g. two papers by Barrett et al that are currently in prep and referenced as such). This overview paper has to strike a balance between presenting key findings and results, while not 'stealing the glory' of all the other papers that are referred to in it. It would be wholly inappropriate if the overvbiew paper became the only go-to reference for all the results in the CALRIFY-2017 campaign. However, we agree that some of the impact of the figure is lost because originally it appeared in conjunction with Figure 9, that shows the discrimination between the various meteorological regimes that were encountered during the measurement campaign. To combat this, we have reproduced the same discrimination bands that were shown in Figure 9 directly on the Figure. We thanks the reviewer in advance for their understanding of the difficult balance it is intended to be an overview paper that signposts the results in other papers.

l517 why "also"? at first glance the two statements (vertical gradient in SSA, vertically constant mass absorption coefficient) are in contradiction.

Also removed. The results are not contradictory if one considers the sentence immediately following: "Wu et al (2020a) and Taylor et al. (2020) propose that the partitioning of a higher fraction of inorganic ammonium nitrate onto the existing particles at the colder temperatures associated with the higher altitudes explains the vertical structure in SSA in the region above Ascension Island." There is more scattering nitrate, reducing the relative contribution of absorption and hence increasing the SSA.

l598 isn't this contradicting the result quoted earlier that SSA increased with height?

We don't think so. We couldn't find exactly what the point being made was here.

l564-617 In my opinion, this is a bit of a lengthy discussion of some details that maybe is not a headline result of CLARIFY (i.e. this section might be shortened).

We think that a headline result from CLARIFY-2017 is that we are using the most accurate absorption measurement techniques that are available to the airborne atmospheric aerosol community. The vast majority of measurements of optical properties have been made with vastly inferior filter-based instrumentation which results in a high degree of uncertainty in the single scattering albedo. This fact is not often appreciated by modellers. We therefore believe that it is our duty to highlight them so that modellers understand the difference between these and filter based measurements. A 45% error in absorption from filter based measurements using the Bond correction algorithm means that we would be stuck with unacceptably high uncertainties in any derived aerosol optical depth. It is absolutely a key aim and objective to characterise BBA absorption as well as possible, which justifies our keeping the text as is.

l692 the *refs* are missing ...

Apologies. Three suitable references are now included.

l699-701 the ± ranges are awkward, as they imply strong possibility of negative values. An asymmetric range is required in my opinion.

Agreed. We now present the inter-quartile ranges.

l703 μ; are the median values for diameters as in Fig. 16? this should be clarified as it is uncommon (typically radii are reported).

Agreed. We have swapped to cloud effective radius.

l706 why is this result not clear? Isn't this a large factor? It is also consistent with the results reported for LWP and for Zct.

Agreed that the phrase was not helpful. We have reworded this.

l774-786 This seems to me a pure modelling study that is disconnected to the observations of CLARIFY. If so, I suggest to drop the discussion. If my impression is wrong, it would be useful to make the link to the observations at this point.

We don't think that the study is disconnected. After all some of the rationale figures (e.g. Fig 2, Fig 3) are entirely model based. We therefore prefer to leave this in the text.

l816-819 I don't remember a detailed discussion of this result in the main body. Is it a key CLARIFY observational result?

We believe that the words in questions are these: "Mie scattering theory using simple mixing rules such as volume weighting of refractive indices, or the Maxwell-Garnet mixing rule are not able to simultaneously represent both the mass absorption coefficient and the SSA of the BBA (Taylor et al., 2020). This has implications for how to represent aerosol optical properties in global climate models that are fully consistent between the chemical and optical properties."

We already explicitly state this, which we do think adequately represents the findings (the text is modified a little to emphasize this important finding more clearly:-

"Taylor et al (2020) show that aerosol optical properties are not well represented when using Mie scattering theory and volume weighting of refractive indices that is currently used in many GCMs; such simple mixing rules are unable to simultaneously represent both the mass absorption coefficient and the SSA of the BBA. While the models documented in Table 3 utilise volume weighting of refractive indices, the resultant mass absorption coefficient (i.e. the mass-normalised aerosol absorption cross section) using a straightforward Mie theory model with these volume-weighted effective refractive indices does not agree with measurements derived from the EXSCALABAR, AMS and SP2 instruments (Taylor et al., 2020). Internally consistent optical closure of both the optical parameters and the mass absorption coefficient can be improved using core-shell Mie scattering treatment of a black carbon core and an organic/inorganic coating but can be most accurately reconciled using more complex semi-empirical parameterisations of mixing state (Taylor et al., 2020)."

l820-824 It would be useful to make the connection to the observations, i.e. that in only a quarter of the cases, a SDE is expected.

Given the disagreement between the LES (Herbert et al., 2020) and larger scale models (Che et al., 2020a), we feel it imprudent to state quantitative figures as the results between the two studies appear contradictory as stated in the text.

l829-831 This seems to be a bit out of the blue, in particular the statement on the indirect effect, especially given the peculiar indirect effect in GLOMAP. A more clear link to the body text and the observations is necessary.

Agreed. This conclusion was not adequately supported in the main text. We have rephrased this to focus on the direct effect and the results that are shown in Fig 18:-

"Simulations with the UMESM1 climate model performed at different spatial resolutions are represent the aerosol direct effect consistently across model resolutions, which shows the advantages of the Unified Model framework in which the underlying physics is identical between high resolution and lower resolution simulations."

l834-837 Sorry if again I missed a point, but were aerosol-radiation interactions from MODIS shown in the body, and compared to the SEVIRI results?

In the body of the text we state this: "Peers *et al*. (2019, 2020), has developed a novel above cloud aerosol detection algorithm from the geostationary SEVIRI instrument. Importantly, this retrieval accounts for the impacts of water vapour in the relatively wide SEVIRI spectral bands by assimilating humidity profiles from the Met Office NWP global model leading to improvements in the accuracy of the retrievals (Chang and Christopher, 2016). A comparison against above-cloud retrieval algorithms developed from MODIS (Meyer *et al.*, 2015) has been performed revealing some systematic differences, but overall the agreement in cloud and aerosol properties is satisfactory (Peers *et al*., 2020). The geostationary nature of the SEVIRI satellite instrument means that, unlike polar orbiting

satellite retrievals which require precise colocation, coherent comparisons between aircraft and SEVIRI retrievals are possible. A number of cases have been investigated, with encouraging agreement between aircraft and SEVIRI retrievals (Peers *et al.*, 2020)."

We therefore believe that we have signposted the agreement between MODIS and SEVIRI retrievals of above cloud AOD. However, the reviewer is correct that this does not (quite) get to the aerosol-radiation-interaction estimates of radiative effect. This work is ongoing (and very promising!). Hence we rephrase the conclusions to:-

Despite the relatively broad wavebands used by the SEVIRI geostationary sensor, above-cloud aerosol retrievals derived from a newly developed algorithm were shown to compare favourably to those derived from MODIS provided that water vapour profiles were adequately accounted for (Peers *et al.*, 2019, 2020). The geostationary nature of SEVIRI means that the full diurnal cycle of aerosol radiative effects can be examined with implications for future studies on aerosol-radiation-interactions.

l853 "are thanked"

Changed.

l1206/Table 1 is "resolved number" and "size resolved number" the same or something different (if the former, give it the same name, if the latter, explain). some µ don't appear in the PDF.

Thanks for spotting this. We've made some efforts to homogenise the table.

l1245 it would be useful to show zero COD as white or grey in panel (b) (presumably white in (c) means no cloud?). Why does the figure legend write "Carb."?

We now add that white corresponds to no cloud in the caption in c. From an operational perspective, we preferred plotting it this way as c provides information on cloudy/cloud-free regions.

Regarding 'carb' – we explicitly state the assumptions in the text: "The three components that were chosen were i) sulfate with emissions from industrial pollution and dimethyl sulphide (DMS), ii) a simplified two-bin mineral dust scheme based on Woodward (2001) with interactive emissions and data assimilation from MODIS Aqua, and iii) 'carbonaceous aerosols' with real-time fire emissions from fossil fuel, biofuel and real-time fire emissions (Global Fire Assimilation System 250 (GFAS); Kaiser et al., 2012) combined into one tracer." However, we agree to add explicitly what "carb" represents in the caption.

l1265 the "each year different colour" is probably a remnant from a former version of the Figure? "ARM AOD" or rather "Aeronet AOD"?

Apologies. The reviewer is correct that this was from a previous version. We have amended the caption accordingly.

l1290 The caption for (b) should briefly explain how the mean altitude is calculated/defined.

OK – we have now described this briefly in the caption.

l1295 why not focus on the lower 3000 m in panel (a) so that the differences between radiosondes and model are better visible?

Done.

l1305 The caption needs to clarify where the data is from.

Reference to the work by Taylor et al (2020) and Wu et al (2020) is now included.

l1330 why not also a Gaussian fit to the precipitating days?

There was an important typo that has been corrected in the caption. The data from the non-precipitating days is not fitted with a single Gaussian because of the extended tail in the CO pollution data.

**Reviewer #2.**

We are pleased that the reviewer found the paper very clear and well written, and have incorporated our responses to the reviewer's comments as documented below:-

L49: "Aerosol-cloud interactions, also termed indirect effects, arise from aerosols acting as cloud condensation nuclei (CCN) in warm clouds": Aren't ice clouds concerned too? Should not this be mentionned as well here on the basis of the recent ICE-D detatchment made from Cape Verde?

The focus of the work is very much on the interaction of aerosols with warm clouds via the role as CCN. However, the reviewer is right that aerosols' roles as ice nuclei are also important. As this is rather tangential to the study we amend the sentence to state:-

"Aerosol-cloud interactions, also termed indirect effects, arise from aerosols acting as cloud condensation nuclei (CCN) or ice nuclei in clouds." This sentences therefore acknowledges that aerosols as IN may a role in ice clouds.

L156-157: "However, it is difficult to fully discern the level of interaction between clouds and aerosols because of the sensitivity of lidars in the free troposphere (Watson-Parris et al., 2018) and the attenuating effects of a thick layer of aerosols overlying clouds." This is agreed, but the work by Daeconu et al. (this special issue) has shown that this can be managed as long as one uses CALIOP observations at 1064 nm. I think it would be worth mentionning this study:

Very happy to include reference to this work although it would be more logical to add this later on in the text rather than here. We add this around line 678 of the revised document:

"which owing to the longer wavelength has been shown to be able to penetrate through the thick BBA layers over the S.E. Atlantic (Deconu et al., 2019)"

368: SLR already defined.

Agreed – removed.

Figure 7: could be improved by showing the type of flying as a function of the objectives. . .? And show the kind of vertical sampling made during the flights as well.

It is difficult to show the objectives of the flights on a geographic plot such as this. This is frequently because some parts of the flights are dedicated to e.g. in-situ aerosol sampling, in-situ cloud sampling, radiative transfer Z patterns etc. It is probably best to stick with what we have; a geographic plot and a table that documents the primary objectives for each of the flights. The geographic plot, is interesting in itself because it shows that flights were performed in any direction from Ascension Island rather than in the direction of the climatological mean in the AOD.

With respect to the vertical sampling, we thought this was a good idea as it would allow the reader to assess roughly how much time was spent assessing various different aspects e.g. the MBL and the residual CBL. We have therefore included a plot which details this as a simple pdf plot of the operating altitude of the aircraft and a sentence in the text referring to this:

"An analysis of the time the aircraft spent at different altitudes is shown in Figure 7b, which reveals that 45% of the time the FAAM aircraft was operating in the MBL, and 36% of the time in the residual CBL in the BBA layer, with the remaining time spent above the BBA while transiting to or from mainland Africa (C041, C041) or making radiometric measurements."

Figure 10: It would be nice to show winds associated with the composited geopotential patterns.

Agreed. We have now added the wind vectors as suggested by the reviewer and a few additional word in the text.

Figure 11: nice but why have you not composited the back trajectories on the same days as for the G1 and G2 ? why do this on the whole of August?

We decided to do it this way as, although there are dominant regimes, there are other flows that do not necessarily fall in either category – by plotting the whole of the month we include transitional times when the flows switch from one regime to the other. We are plotting a trajectory density plot, so one can immediately identify the relative importance of the flow from the different directions.